# Prognostic Exploration of U-F-Au-Mo-W Younger Granites for Geochemical Pathfinders, Genetic Affiliations, and Tectonic Setting in El-Erediya-El-Missikat Province, Eastern Desert, Egypt

Mahmoud M. Hassaan [1,*], Sayed A. Omar [2], Ahmed E. Khalil [3], Taher M. Shahin [1], Islam M. El-Naggar [1], M. I. Sayyed [4,5] and Mohamed Y. Hanfi [2,6,*]

1   Department of Geology, Faculty of Science, Al-Azhar University, Cairo 11884, Egypt; taher_gc@azhar.edu.eg (T.M.S.); islam-elnaggar@azhar.edu.eg (I.M.E.-N.)
2   Nuclear Materials Authority, P.O. Box 530, El Maadi, Cairo 11936, Egypt; sayedomar87@yahoo.com
3   Department of Geological Sciences, National Research Centre, El-Buhouth St., Dokki, Cairo 12622, Egypt; ahmedekhalil58@yahoo.com
4   Department of Physics, Faculty of Science, Isra University, Amman 11622, Jordan; mabualssayed@ut.edu.sa
5   Department of Nuclear Medicine Research, Institute for Research and Medical Consultations (IRMC), Imam Abdulrahman Bin Faisal University (IAU), P.O. Box 1982, Dammam 31441, Saudi Arabia
6   Institute of Physics and Technology, Ural Federal University, St. Mira, 19, 620002 Yekaterinburg, Russia
*   Correspondence: mah_hassaan.201@azhar.edu.eg (M.M.H.); mokhamed.khanfi@urfu.ru (M.Y.H.)

**Abstract:** Younger granite bodies form two arches, the western and the eastern (WA, EA), which extend from the south northwards from the Meatique, ophiolitic group-island arc rocks, to the large older granite outcrop to the north. This paper concerns the feasibility of exploration in the El-Erediya-Ria El-Garah-El-Gidami-El-Missikat Y Gr regions. Fieldwork and remote sensing, together with geochemical, petrochemical, and mineralogical studies, are used to show the controlling factors, routes, and the origins of the deposits. Remote sensing is used to delineate the different rock units. Normal and strike–slip NW, NNE faults, veins, fractured ENE shear zones, and alteration zones of magmatic-hydrothermal fluids are discussed. Granites are considered using petrochemical diagrams as resources. These rocks are categorized as syeno- and alkali feldspar granites. Geochemical binary relationships recognized the granites are highly fractionated calc-alkaline-altered Monzo-, syeno-, and alkali feldspar granites formed in the active continental margin. The observed positive Ga vs. Cu, Zn, and Ni correlations are used for epithermal-magmatic-hydrothermal polymetallic veins and mineralized greisen zones. Negative Cu vs. Mo correlation patterns show probable Mo-porphyry deposits in the deeper zones at the contact point between porphyritic perthite and perthite granitic El-Erediya mass. The Zr/Sr between 1.65 to 2.93 plus fluorites in El-Missikat and up to 5.48 plus fluorites in El-Erediya show both U-poor at El-Missikat and U-rich deposits at El-Erediya. The recorded U, Th, Cu, and Pb vertical zoning sequence of deposition differentiates U aureole and deposit zones. The estimated lateral zoning sequences of deposition of these elements define the centers of U deposits. Pathfinders for the deposit of the examined area include the positive $Fe_2O_3$ vs. $MgO$ and $Fe_2O_3$ vs. $CaO$ correlations, and also negative Rb/Sr vs. K/Na and Rb vs. Sr ones, can be applied to future prospecting for similar U-F-Au-W-Mo deposits in the Eastern Desert of Egypt.

**Keywords:** geochemical; exploration; U-F-Au mineralization; eastern desert

## 1. Introduction

The eastern desert of Egypt exposes the Late Proterozoic igneous and metamorphic basement complexes, which are set in the northern section of the Nubian shield that stretches from Somalia to Ethiopia via Sudan. The tectonic Nubian shield is the whole process of canonization of ocean arc complexes, collision and welding to the older African craton, or includes the entire depositional and thermo-tectonic evolution of the crust in

northeast Africa during the time period c. 1200-c. 450 Ma. The Nubian shield in its early stage consisted of a passive continental margin and a back-arc environment. However, its three domains, southern (from the Egyptian border line to Idfu-Marsa Alam road), central (between the Idfu-Marsa Alam and Qena-Safaga asphaltic roads), and northern (north Qena-Safaga road), show three different stages. These three domains are separated by Idfu-Marsa Alam and Qena-Safaga tectonic discontinuity. The southern and central domains are distinguished by ophiolitic, island arc, calc-alkaline volcanic, volcano-sedimentary rocks, and metagabbro. This stage is concluded by the ophiolite sequence of the Meatique group located 10 km north of the Quseir-Qift road and Atalla shear zone. The cordilleran stage is represented by older (tonalite–granodiorite) and younger (Monzo, syeno, and alkali feldspar) granites.

Basement exposures in the eastern desert (ED) are characterized by a diverse variety of granitic rocks [1]. Egyptian granitic rocks are classified according to their apparent relation to orogeny into the syn- to late-orogenic granites (610–880 Ma) and the post-orogenic to anorogenic granites (475–600 Ma). Post-orogenic to anorogenic granites are recognized as older granitoids and younger granites [2]. Nubian Shield (NS) magmatism progresses from arc-related tholeiite and calc-alkaline tonalite-trondhjemite–granodiorite (older granites) assemblage to collisional-related calc-alkaline granites assemblage (younger granites. Finally, the post-collisional within-plate A-type granite (riebeckite granite) is generated during orogenic collapse [3]. The younger granites of the central–northern domains are distinguished as they hold several metallic deposits, chiefly Au, Ag, U, F, Sn, W, and Mo. Such deposits are found in a younger granite belt extending from El- Erediya and El-Missikat in the south to Gebel Gattar and Wadi Elfaliq in the north (Figure 1).

These younger granite (Y Gr) exposures are surrounded by the large older granite (O Gr) located within the district of the Safaga-Qena tectonic discontinuity (TD) (Figures 1b and 2; [4]). The determined age for the younger granite (Y Gr) is 595 Ma–605 Ma, and for the older granite (O Gr) is 652 ± 2.6 Ma [5]. Meanwhile, the reported ages for both granites by [6] are 590 Ma–610 Ma and 620 Ma, respectively. The Y Gr bodies are classified based on the alkali-feldspar/total feldspar ratio on QAPF diagram of [7] into monzogranite (0.35–0.65), syenogranite (0.65–0.90), and alkali feldspar granite (more than 0.90). They are also distinguished by Ca content of less than 1%.

The El-Erediya, Ria El-Garrah, El-Gidami, and El-Missikat Y Gr plutons are bearing uranium and fluorite deposits, scattered greisen zones bearing uranium-fluorite-sulphide-gold and minor fluorite and polymetallic veins (minerals of U, W, Mo, Pb, As, Cu, less anomalous Ag-Au [8–14]).

The present study aimed to study the geochemical characteristics and pathfinder elements for the studied granites and determine whether these deposits originated from hydrothermal fluids around their hosting magmatic centers. This study used petrochemical, geochemical, and remote sensing data to reveal pathfinders that will also be helpful to explore other such U- F- Au Y Gr geochemical provinces in the eastern desert (ED) of Egypt.

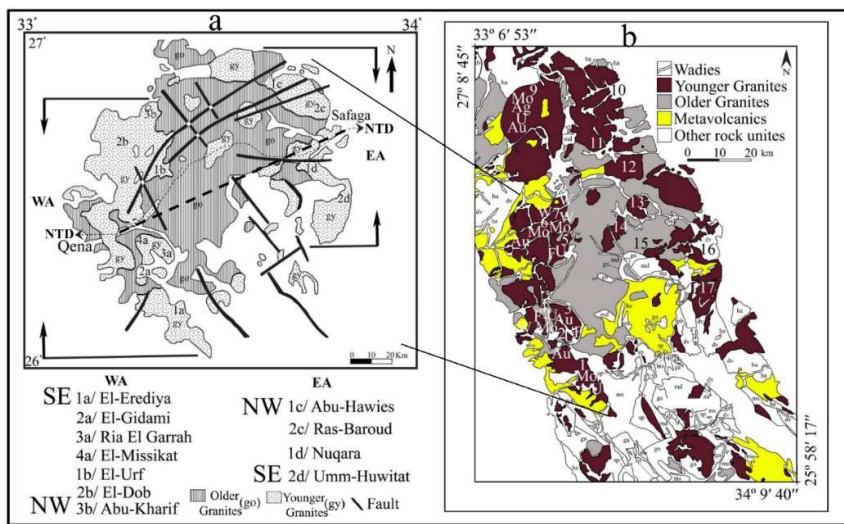

**Figure 1.** (**a**) Location map of the western (WA) and eastern (EA) arches of younger Granite (Y Gr.) surrounding huge older Granite (O Gr.), District of the Northern Tectonic Discontinuity (NTD), the [4]. (**b**) The granitic outcrops of the two convex arches modified after [15] are (1) G. El-Eradiya, (2) G. El-Gidami, (3) G. Ria El- Garrah, (4) G. El-Missikat, (5) G. El Urf, (6) G. El Dob, (7) W. El Dob, (8) G. Abu Kharif, (9) G. Qattar, (10) W. Faliq el-Wair, (11) G. Shayib, (12) G. Umm Anab, (13) G. Ras Barud, (14) G. Abu Hawis, (15) G. Abu Furad, (16) G. Nuqqara, (17) G. Umm-El Huwitat, Gabbro (ga), Post-Hammamat Felsite (fph), Hammamat group (ha), Dokhan Volcanics (dv), Metagabbro-Diorite complex (md), Serpentinite (sp), Metasediments (ms), and Paragneisses and Migmatites (gn).

## 2. Materials and Methods

In this investigation, the integration of fieldwork and Landsat-8-ASTER (Advanced Spaceborne Thermal Emission and Reflection Radiometer) data analysis was used to recognize rock units, alterations types, and structural elements (lineaments, faults, shear zones). Thirty-six samples which represent the studied granitic rocks were collected, of which fifteen samples were analyzed using X-ray fluorescence analysis for the major oxides and the trace element concentration. The analysis was conducted using the XRF Technique (Axios Sequential apparatus WD XRF Spectrometer, Philips-P Analytical 2005) and the ASTM-E1621 standard guide for elements analysis by wavelength dispersive X-ray fluorescence spectrometer. The ASTM D7348 standard test method for Loss on Ignition (L.O.I.) of solid combustion was used as a guideline of the model at the laboratories of the National Research Center, Gizza, Egypt.

The techniques used in the present study are remote sensing-based analysis, Landsat-8 (scene path 174, row 42) acquired on 24 June 2021, and ASTER Level-1B, date 11 November 2005, with radiometric and geometric correction coefficients, projected to UTM, Zone 36 North, and datum WGS-84. Digital image processing was also applied using ENVI 5.2 and Arc GIS (10.3) software programs. The processing methods include false-color composite (FCC), band ratio (BR), and principal component analysis (PCA).

In addition, the processing of the averages of 170 major oxide analyses of 29 Y Gr bodies of the central domain of the eastern desert [16] was used to recognize their general geochemical characteristics. Geochemical data of $Al_2O_3$, $Fe_2O_3$, CaO, MgO, $Na_2O$, $K_2O$, Sr, Rb, cU, eU, eTh, and some of the trace elements of 92 chemical analyses given in previous studies, as well as 15 analyses of the collected samples, were used to reveal pathfinders for the studied plutons. The content analyses of cU, eU, and eTh were carried out at the NMA Laboratories using the RS-230 BGO-Super spectrometer and calibration of the RS-230 in equivalent uranium (eU) and thorium (eTh). The cU content was estimated using the U-Laser Analyzer technique. The chemical data of the trace elements of all analyses were used to calculate the Clark of concentration (CC), the correlation coefficient (r), and the

Zoning Coefficient of deposition of elements (γ). These factors are considered the most reliable for geochemical prospecting.

*Geological Setting*

The study area, determined using the field observations, the geologic, and metallogenic maps, is occupied mainly by Neoproterozoic ophiolitic and island arc rocks. These rocks are related to the oceanic terrain of the northwestern Nubian Shield that was collided and amalgamed in the Central Eastern Desert (arc-arc accretion). These rocks were followed by cordilleran syn, late, and post magmatic granites.

The Meatique ophiolitic Group, island arc metamorphic volcanic, volcano-sedimentary, gabbroic rocks, and older and younger granites (Figure 1b; [15]) are arranged from south northwards. This matches the dating of ophiolitic-gabbro, Type I granitoids and metamorphic garnets, suggesting that rifting was active at c. 750 Ma, subduction between c. 760 and 650 Ma, and basin closure. This also suggests that ophiolite obduction, uplift of Meatique Dome, and metamorphism had taken place by c. 700 Ma [17,18].

The granitic rocks are represented by older granites (O Gr) and younger granites (Y Gr), covering most parts of the study area. The Y Gr plutonic outcrops are tectonically intruded and distributed surrounding large O Gr bodies forming two huge regional arches; western (WA) and eastern (EA). Both arches are dissected by Safaga-Qena tectonic discontinuity. The studied El-Erediya, Ria El-Garrah, El-Gidami, and El-Missikat southern Y Gr bodies with Gebel El-Urf, El-Dob, and Abu-Kharif northern Y Gr plutons constitute the western arch (Figure 1a).

The older granitoids are tonalite-granodiorite, gneissose granodiorite, and quartz diorite, which almost cover the northwestern and eastern parts of Gabal El-Missikat (Figure 2).

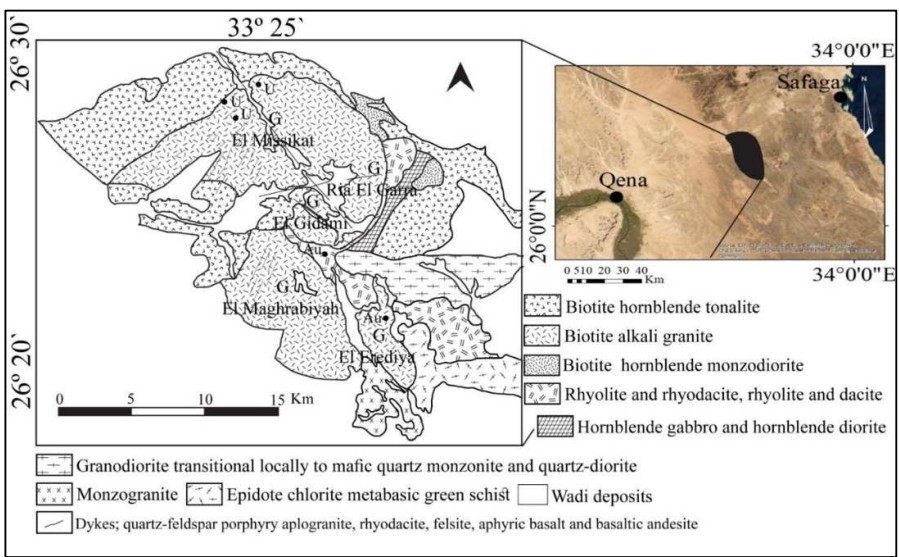

**Figure 2.** Geologic map of El-Missikat, Ria El Garrah, El-Gidami, and El-Erediya granite outcrops bearing U-F-Au deposits [19].

The O Gr rock units are medium to coarse-grained, greyish in color, with predominant xenoliths and alignment of mafic minerals in the form of gneissose texture at contact with Y Gr. They are highly jointed and cross-cut with sharp contact with the alkali feldspar granite of Gebel El-Missikat (Figure 3A). The O Gr body is invaded directly by the amphibolites and metavolcanic rocks.

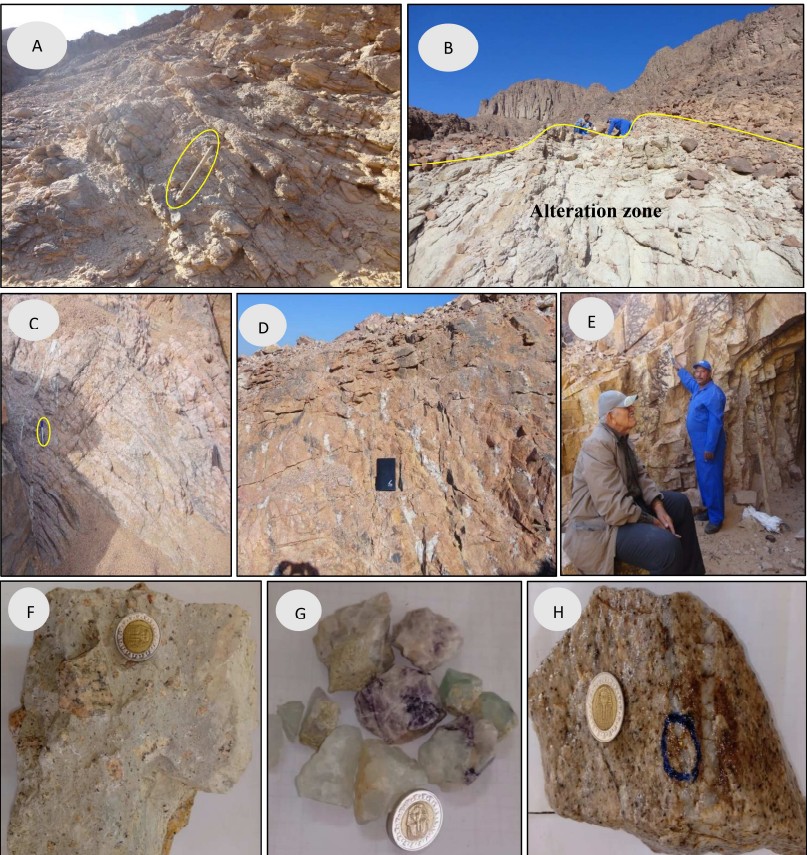

**Figure 3.** Field photographs showing (**A**) highly jointed granite strike (NW)-dip (SW), looking (WS) at Wadi El- Missikat. (**B**) A view of argillic and phyllic alteration in granitic rocks from Gebel El-Missikat. (**C**) Silicification and argillic alteration in granites at Geble El- Erediya. (**D**) Stockwork of quartz veins invaded in the altered granite. (**E**) Old shaft of mining area in the alteration zone of the alkali feldspar granite. A hand sample of (**F**) argillic alteration. (**G**) Fluorite. (**H**) Sulphides in quartz micro-veinlets.

The younger granites form high mountains, including G. El-Erediya, G. Ria El-Garrah, and G. El-Gidami, covering almost all of the study area. They are classified into perthitic leucogranite [12] and alkali feldspar granite [13] in El-Missikat and El-Erediya. They are characterized by massive, medium-grained, pink to red color and are dissected by veins and a network of quartz and aplite of post granitic dykes (Figure 3C,D). They are jointed and weathered; exfoliation and onion-like shapes are common. The U-F-Au deposits in the study area are recorded along a major shear zone (Figure 3E,G) with jasperoid veins occupying faults and fractures trending NW of the Najd fault system [14].

A change in the tectonic regime from compression to extension is associated with U-F-Au deposits in the eastern desert. These deposits occurred at the end of a supercontinental cycle that included the breakup of Rodinia and the development of Gondwana [3]. This cycle had several stages, including the development of the Mozambique Ocean, volcanic arcs, and fore-and back-arc ophiolites, followed by ocean closure, arc accretion terranes by subduction, continental collision (East African orogeny), and a transition from calc-alkaline to alkaline granitoid magmatism.

## 3. Results and Discussion

### 3.1. Remote Sensing Data Analysis

The remote sensing digital image processing techniques for the Landsat-8 and ASTER satellite images, including false-color composite (FCC), band ratio, and principal components analysis (PCA) methods supported by [19], (Figure 2) were used to discriminate the rock unites and map spectral signatures associated with the hydrothermal alterations in

the studied area. The fieldwork and FCC image in RGB (Figure 4) delineated the rock units, and PCA recognized the alteration minerals (sericite, illite, and chlorite). The Y Gr bodies appeared in a light blue color, the O Gr body in a violet color, and the metavolcanic rocks in a yellowish color (Figure 4).

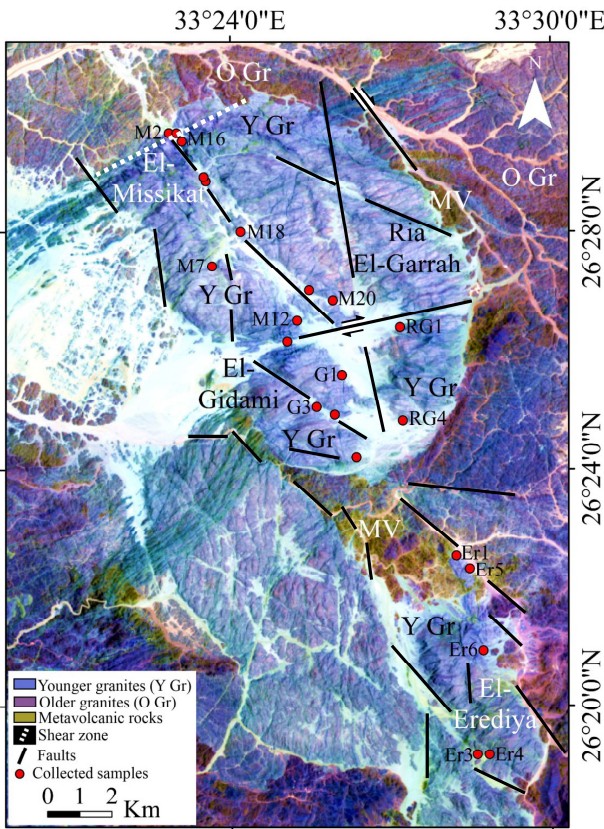

**Figure 4.** False-color composite (FCC) Landsat-8 (2, 4, and 7 in RGB, respectively) image, discriminating older and younger granites and showing the extracted faults.

### 3.1.1. Structures

Three main deformation events (D1–D3) were found in the eastern desert [20,21]. D1 is relevant to high-grade infracrustal rocks, represented by an isoclinal fold in the amphibolite body and gneisses core complexes. D2 and D3 are related to supracrustal ophiolite and arc-related rocks comprised in the study district Meatiq ophiolitic group and island metagabbro. D2 is interpreted as an accretion and subduction stage and is represented by compressional and transpressional-related thrust, ductile shear zones and folds with E and NE directions often formed the El-Erediya and El-Missikat shear zones. D3 is recognized as a collision stage by strike–slip faults of dextral and sinistral types of NW-NNW trends, as well as brittle–ductile shear zones and normal faults of N trend cutting El-Gidami-El-Garrah.

The applied remote sensing techniques were verified during field study and revealed lineaments, faults, and shearing. The extraction of lineaments results from using the Line Module in PCI Geomatica, which is the most widely applied software [22]. This is reached by combining isolated eight shaded light lineament images of eight-azimuth angles 45°, 90°, 135°, 180°, 225°, 270°, and 315° of the light source. The lineament map shows that the studied granites were subjected to intense deformation processes (Figure 5).

The recognized faults are trending NW–SE, ENE–WSW, and N–S (Figure 4). Two strike–slip faults trending ENE–WSW and NW–SE, at El-Missikat and El-Erediya respectively, are cut with displacement. The ENE–WSW strike–slip fault separates Ria El Garra's southern part and El Gidami from El-Missikat. The El-Erediya NW strike–slip fault, cut

with displacement, a fault trending NNE which is situated within the porphyritic perthite granite central part.

The rose diagrams for the lineaments using the rock works software detected certain main trends and azimuth directions for each granite body (Figure 5a–d) as follows; El-Erediya granite: NW–SE chief trend and NE–SW minor trend. Ria El-Garra: ENE–WSW chief trend and NW–SE minor trend. El Gidami: ENE–WSW and NW–SE chief trends. El-Missikat granite: ENE–WSW chief trend and WNW–ESE minor trend

A reactivated fractured shearing within several NW faulted zones dissects the northwestern margins of the El-Missikat body at its contact with the gneissose quartz diorite (Figure 3A). This shear zone trending ENE–WSW to NE–SW, and dipping about 60°–70° to SE until the zone of the ENE–WSW strike–slip fault which separates both the Ria El-Garra's and the El Gidami from the El-missikat and the Ria El Garra's northern part (Figures 4 and 5a–d). The ENE trend apparently represents reactivated tensional fractures genetically related to the ENE left-lateral strike–slip fault [23]. This shear zone hosts polymetallic veins and scattered greisen zones bearing coarse-grained sulfide-gold, W, Mo, Pb, As, Cu, and less anomalous Ag-Au beside the uranium and fine- to coarse-grained fluorite deposits [8,14].

### 3.1.2. Alteration Types

The band ratio (BR) and principal component analysis (PCA) techniques are more useful for mapping the alteration types. ASTER band ratios of $4/(5 + 6)$ and $(4/6)$ after [24] were used to detect argillic and phyllic (Figure 6) alteration that manifested as red and green colors, respectively. Propylitic alteration was also recognized by the $(4/2–4/5–5/6)$ ratio of [25], which is represented by a light blue color (Figure 6).

The principal component analysis (PCA), using [26] technique, determined the best four bands for each alteration mineral: two for maximum reflectance and the other two for absorption features. According to the alteration products found in the study area through the field, petrographic, and previous works, the following common end member alteration minerals were selected: sericite, illite, and chlorite (Figure 7). The argillic alterations usually consisted of illite, kaolinite, smectite, the phyllic of sericite, pyrite, calcite, K-feldspar, kaolinite, secondary quartz, and biotite and the propylitic of epidote, chlorite, Na-feldspar [27].

The following are the microscopic discovered alteration minerals: secondary quartz, epidote, muscovite, sericite, chlorite, iron oxides, goethite (after pyrite), hematite, and calcite were found at El-Erediya-El-Missikat [9,13,14,28]. Sericite, chlorite, iron oxides, and muscovite were among the minerals found at Ria El-Garrah-El-Gdami [10,11].

The four granite bodies were subjected to the following alteration processes: Argillic (100–300 °C) showed biotite, chlorite, alanite, sulfides, quartz, illite, kaolinite, and andalusite minerals, locally developed at relatively shallow levels of the hydrothermal system (Figure 6a,d,g,j). Propylitic (250–400 °C) represented epidote, chlorite, calcite, pyrite, iron oxides, sericite, and apatite minerals, extensively developed around most porphyry deposits extended up to El-Missikat (Figure 6c,f,i,l). Phyllic (200–450 °C), characterized by sericite, pyrite, K-feldspar, kaolinite, secondary quartz, calcite, and biotite minerals, was associated with a relatively later stage of the development of the hydrothermal system (Figure 6b,e,h,k). The decrease of the effect of each argillic, propylitic, and phyllic process followed the decrease of the temperature of formation of each from El-Erediya at the south to the northwestern boundary of El-Missikat via El-Gidami and Ria El-Garra (Figures 5–7). The revealed alterations within the erosional crust of the four plutons favor that the erosional crust is the upper level of each mineral deposit.

The W-Mo-Au-F-U deposits in the examined granites are hydrothermal deposits generated around magmatic (co-magmatic) centers, according to the [27] typical alteration processes. The core porphyritic perthite granite section of El Erediya granite suggests the presence of a Mo porphyry deposit in this regard. Due to sodic alteration, the northwestern mass of El-Erediya granite is albite granite, according to [9]. Sodic-potassic alteration

processes produced the center porphyritic perthite and southeastern perthite granite groups. Of the three components, hydrothermal silicification is the most common.

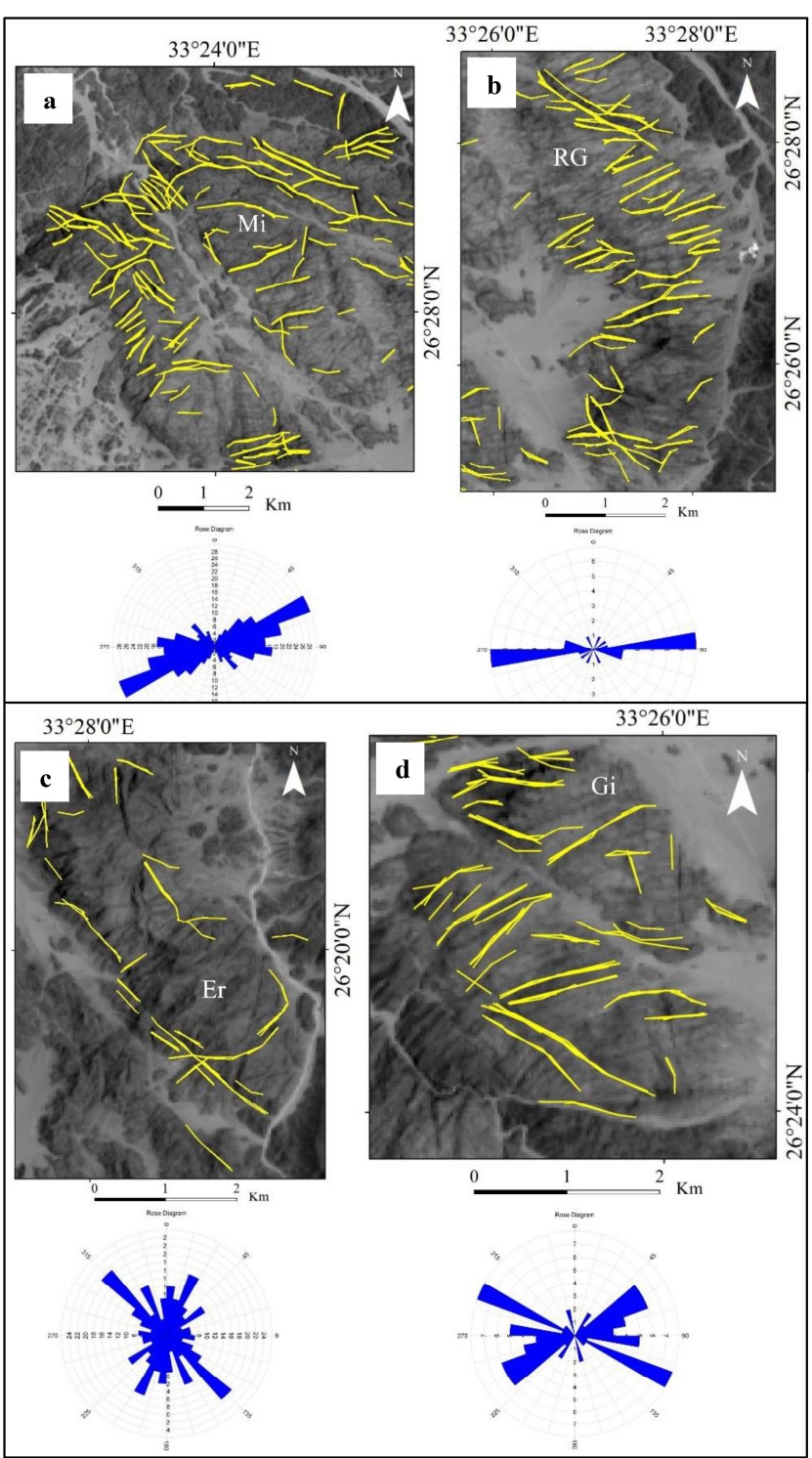

**Figure 5.** Lineaments of the studied granites outcrops with rose diagrams ASTER image; (**c**) symbol Er = El-Erediya, (**d**) Gi = El-Gidami, (**a**) Mi = El-Missikat, and (**b**) RG = Ria El-Garrah.

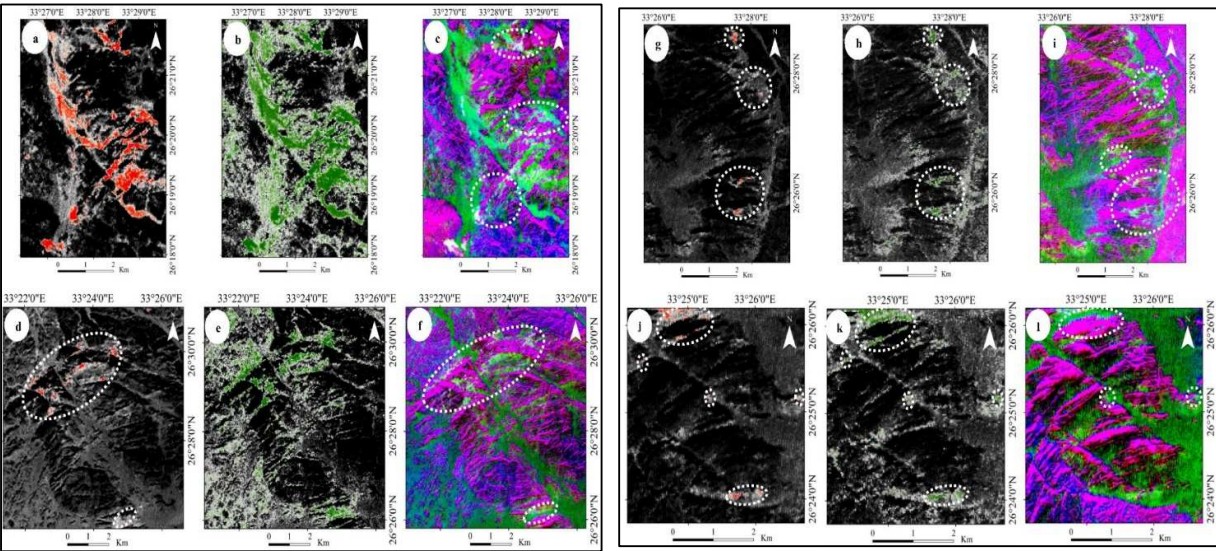

**Figure 6.** Aster band ratios (BR) 4/(5 + 6) [24], (4/2–4/5–5/6) [25], and 4/6 [24], showing Argillic in (**a**) El-Erediya, (**d**) El-Missikat, (**g**) Ria El-Garrah and (**j**) El-Gidami), Propylitic in (**c**) El-Erediya, (**f**) El-Missikat, (**i**) Ria El-Garrah and (**l**) El-Gidami, and Phyllic in (**b**) El-Erediya, (**e**) El-Missikat, (**h**) Ria El-Garrah and (**k**) El-Gidami) alterations.

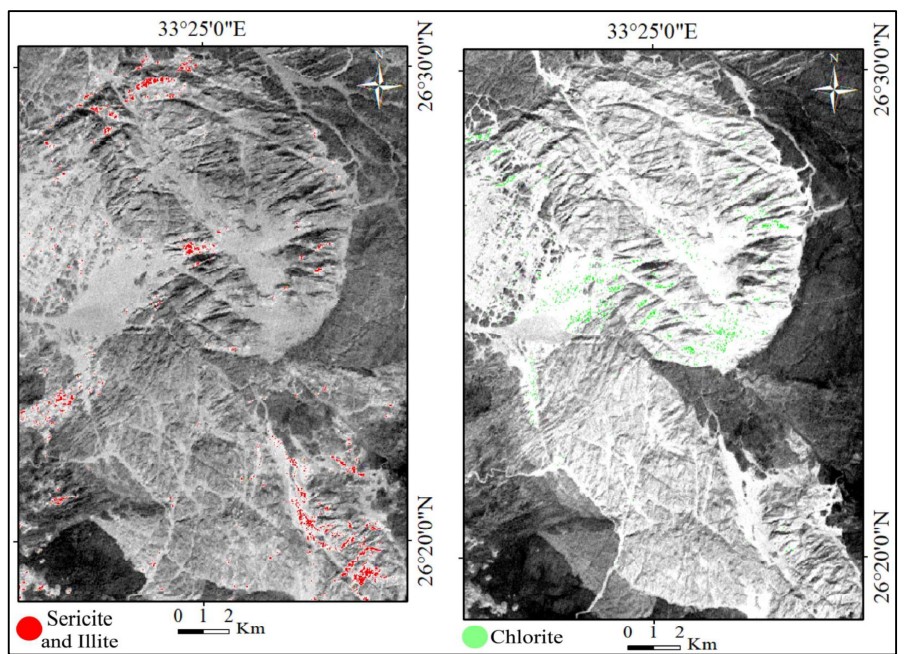

**Figure 7.** Aster PC3 image showing sericite and lllite, chlorite and epidote alteration minerals.

*3.2. Geochemical Characterization*

3.2.1. Petrochemical Features

The 15 chemical analyses, Niggli values, and CIPW norm values of the studied granites (Table 1) were used to define the petrochemical types of the four studied plutons.

**Table 1.** Major oxides (wt. %), trace elements (ppm), CIPW and Niggli Norms of the studied granites in El-Missikat-El Erediya.

| Area | El Missikat | | | | | | Riea El Garrah | | El Gidami | | El Erediya | | | | |
|---|---|---|---|---|---|---|---|---|---|---|---|---|---|---|---|
| S. No. | M2 | M7 | M12 | M16 | M18 | M20 | RG1 | RG4 | G1 | G3 | Er1 | Er3 | Er4 | Er5 | Er6 |
| **Major oxides** | | | | | | | | | | | | | | | |
| $SiO_2$ | 73.50 | 72.83 | 73.12 | 71.67 | 69.44 | 72.63 | 74.22 | 69.31 | 72.55 | 73.50 | 71.98 | 72.40 | 72.65 | 72.62 | 71.74 |
| $TiO_2$ | 0.04 | 0.20 | 0.10 | 0.06 | 0.09 | 0.09 | 0.15 | 0.28 | 0.07 | 0.07 | 0.06 | 0.18 | 0.09 | 0.02 | 0.06 |
| $Al_2O_3$ | 14.40 | 13.95 | 14.75 | 15.08 | 16.55 | 14.29 | 13.49 | 14.80 | 14.61 | 14.05 | 15.99 | 14.56 | 14.36 | 14.88 | 14.87 |
| $Fe_2O_3$ | 1.11 | 2.19 | 1.09 | 1.44 | 1.37 | 1.19 | 1.65 | 2.95 | 1.45 | 1.33 | 0.50 | 1.86 | 1.71 | 1.03 | 1.50 |
| $CaO$ | 0.56 | 1.18 | 0.88 | 0.72 | 1.33 | 1.14 | 0.80 | 1.37 | 0.83 | 0.80 | 0.08 | 1.24 | 0.92 | 0.55 | 0.80 |
| $MnO$ | 0.04 | 0.05 | 0.01 | 0.03 | 0.01 | 0.01 | 0.03 | 0.06 | 0.03 | 0.03 | 0.01 | 0.12 | 0.06 | 0.04 | 0.05 |
| $MgO$ | 0.06 | 0.26 | 0.11 | 0.08 | 0.14 | 0.13 | 0.20 | 0.30 | 0.07 | 0.07 | 0.02 | 0.28 | 0.19 | 0.08 | 0.14 |
| $Na_2O$ | 4.68 | 4.10 | 4.44 | 4.56 | 4.69 | 4.08 | 4.25 | 3.92 | 4.25 | 4.29 | 5.24 | 3.88 | 4.72 | 5.50 | 5.37 |
| $K_2O$ | 4.92 | 4.80 | 4.91 | 5.66 | 5.53 | 5.56 | 4.99 | 6.44 | 5.56 | 5.22 | 5.83 | 5.16 | 5.07 | 4.57 | 4.56 |
| $P_2O_5$ | 0.01 | 0.07 | 0.01 | 0.01 | 0.01 | 0.01 | 0.03 | 0.05 | 0.01 | 0.01 | 0.02 | 0.05 | 0.05 | 0.01 | 0.01 |
| $SO_3$ | 0.01 | 0.02 | 0.07 | 0.07 | 0.22 | 0.29 | 0.07 | 0.03 | 0.07 | 0.07 | 0.07 | 0.03 | 0.05 | 0.03 | 0.22 |
| L. O. I. | 0.05 | 0.03 | 0.09 | 0.03 | 0.06 | 0.05 | 0.03 | 0.03 | 0.08 | 0.03 | 0.02 | 0.04 | 0.06 | 0.04 | 0.03 |
| **Total** | 99.36 | 99.66 | 99.58 | 99.41 | 99.42 | 99.45 | 99.91 | 99.52 | 99.57 | 99.48 | 99.80 | 100.43 | 99.93 | 99.37 | 99.34 |
| **Trace elements** | | | | | | | | | | | | | | | |
| Cr | 50 | 15.7 | 6.8 | 5 | 7.8 | 65 | 16.8 | 120 | 16 | 45.9 | 40.1 | 103.8 | 36.1 | 109.8 | 18.5 |
| Ni | 40 | 30 | 40 | 60 | 40 | 30 | 2.6 | 70 | 50 | 60 | 50 | 2.8 | 1.3 | 30 | 50 |
| Cu | 40 | 30 | 40 | 20 | 20 | 0 | 4.6 | Bdl | 30 | 40 | 40 | 5.5 | 4.2 | 50 | 50 |
| Zn | 120 | 210 | 200 | 130 | 120 | 110 | 39.8 | 120 | 360 | 70 | 680 | 42.4 | 26.8 | 450 | 260 |
| Zr | 180 | 240 | 180 | 190 | 220 | 260 | 106.2 | 280 | 170 | 220 | 650 | 105.6 | 66.4 | 280 | 300 |
| Rb | 540 | 230 | 260 | 530 | 350 | 360 | 127.8 | 260 | 330 | 300 | 820 | 233.4 | 167.4 | 610 | 390 |
| Y | 210 | 70 | 130 | 200 | 170 | 170 | 64.4 | 110 | 150 | 170 | 90 | 58.9 | 38.7 | 110 | 230 |
| Ba | 580 | 270 | 100 | 1050 | 450 | 610 | 31 | 280 | 450 | 300 | 140 | 60.1 | 53.4 | 120 | 120 |
| Sr | 10 | 110 | 30 | 20 | 20 | 30 | 13.3 | 80 | 20 | 20 | 40 | 35.3 | 77.7 | 20 | 30 |
| Ga | 50 | 40 | 50 | 60 | 50 | 50 | 22.2 | 40 | 30 | 50 | 110 | 22.1 | 17.7 | 60 | 70 |
| Nb | 80 | 50 | 70 | 110 | 100 | 90 | 17.5 | 60 | 80 | 100 | 80 | 29.1 | 24.1 | 100 | 110 |
| Ti | Bdl | Bdl | Bdl | Bdl | Bdl | Bdl | 193.8 | Bdl | Bdl | Bdl | Bdl | 276.4 | 180.8 | Bdl | 10 |
| **CIPW Norm** | | | | | | | | | | | | | | | |
| Q | 25.5 | 26.7 | 25.9 | 21.03 | 17.3 | 24.8 | 27.8 | 17.6 | 23.9 | 26.1 | 18.8 | 26.1 | 22.9 | 21.3 | 20.5 |
| Or | 29.1 | 28.4 | 29.02 | 33.5 | 32.7 | 32.9 | 29.5 | 38.1 | 32.9 | 30.9 | 34.5 | 30.5 | 29.9 | 27 | 26.9 |
| Ab | 39.6 | 34.7 | 37.6 | 38.6 | 39.7 | 34.5 | 35.9 | 33.2 | 35.9 | 36.3 | 44.3 | 32.8 | 39.9 | 46.5 | 45.4 |
| An | 2.7 | 5.4 | 4.3 | 3.5 | 6.5 | 4.3 | 2.9 | 3.8 | 4.1 | 3.7 | 0.3 | 5.8 | 3.01 | 2.4 | 2.9 |
| C | 0.4 | 0.03 | 0.6 | 0.2 | 0.5 | - | - | - | 0.1 | - | 0.9 | 0.5 | - | - | - |
| Ap | 0.02 | 0.2 | 0.03 | 0.03 | 0.03 | 0.02 | 0.1 | 0.1 | 0.02 | 0.03 | 0.1 | 0.1 | 0.1 | 0.02 | 0.03 |
| Py | 0.01 | 0.02 | 0.1 | 0.1 | 0.2 | 0.2 | 0.1 | 0.02 | 0.1 | 0.1 | 0.1 | 0.02 | 0.04 | 0.02 | 0.2 |
| Il | 0.08 | 0.4 | 0.2 | 0.1 | 0.2 | 0.2 | 0.3 | 0.5 | 0.1 | 0.1 | 0.1 | 0.3 | 0.2 | 0.04 | 0.1 |
| Mt | 0.2 | 0.4 | 0.2 | 0.3 | 0.3 | 0.2 | 0.3 | 0.6 | 0.3 | 0.3 | 0.1 | 0.4 | 0.3 | 0.2 | 0.3 |
| Di | - | - | - | - | - | 1.2 | 0.7 | 2.4 | - | 0.2 | - | - | 1.1 | 0.2 | 0.8 |
| Hy | 1.6 | 3.3 | 1.5 | 2.01 | 1.9 | 0.9 | 2.1 | 3.1 | 1.9 | 1.7 | 0.6 | 3.1 | 2.1 | 1.5 | 1.7 |
| **Total** | 99.2 | 99.5 | 99.4 | 99.2 | 99.1 | 99.1 | 99.7 | 99.3 | 99.3 | 99.3 | 99.7 | 99.6 | 99.7 | 99.2 | 99.06 |

**Table 1.** *Cont.*

| Area | El Missikat | | | | | | Riea El Garrah | | El Gidami | | | El Erediya | | | |
|---|---|---|---|---|---|---|---|---|---|---|---|---|---|---|---|
| | | | | | | **Niggli Norm** | | | | | | | | | |
| Q | 23.8 | 24.9 | 24.1 | 19.5 | 16 | 23.115 | 25.9 | 16.4 | 22.2 | 24.4 | 17.3 | 24.3 | 21.3 | 19.7 | 19.02 |
| Or | 29.2 | 28.6 | 29.2 | 33.6 | 32.7 | 33.115 | 29.7 | 38.4 | 33.04 | 31.1 | 34.1 | 30.7 | 29.9 | 26.9 | 26.9 |
| Ab | 42.2 | 37.2 | 40.1 | 41.1 | 42.1 | 36.933 | 38.4 | 35.5 | 38.4 | 38.9 | 46.6 | 35.1 | 42.4 | 49.3 | 48.2 |
| An | 2.8 | 5.5 | 4.3 | 3.5 | 6.5 | 4.289 | 3.01 | 3.8 | 4.1 | 3.7 | 0.3 | 5.9 | 3.02 | 2.4 | 3 |
| C | 0.4 | 0.03 | 0.6 | 0.2 | 0.5 | - | - | - | 0.1 | - | 1.04 | 0.5 | - | - | - |
| Ap | 0.02 | 0.2 | 0.02 | 0.02 | 0.02 | 0.021 | 0.1 | 0.1 | 0.02 | 0.02 | 0.04 | 0.1 | 0.1 | 0.02 | 0.02 |
| Py | 0.01 | 0.02 | 0.1 | 0.07 | 0.3 | 0.305 | 0.1 | 0.03 | 0.1 | 0.1 | 0.1 | 0.03 | 0.1 | 0.03 | 0.2 |
| Il | 0.06 | 0.3 | 0.14 | 0.08 | 0.1 | 0.126 | 0.2 | 0.4 | 0.1 | 0.1 | 0.1 | 0.3 | 0.1 | 0.03 | 0.1 |
| Mt | 0.2 | 0.3 | 0.2 | 0.2 | 0.2 | 0.164 | 0.2 | 0.4 | 0.2 | 0.2 | 0.1 | 0.3 | 0.2 | 0.1 | 0.2 |
| Di | - | - | - | - | - | 1.078 | 0.6 | 2.2 | - | 0.2 | - | - | 0.9 | 0.2 | 0.7 |
| Hy | 1.4 | 2.9 | 1.4 | 1.8 | 1.7 | 0.854 | 1.9 | 2.8 | 1.7 | 1.5 | 0.5 | 2.8 | 1.9 | 1.3 | 1.6 |
| Total | 100 | 100 | 100 | 100 | 100 | 100 | 100 | 100 | 100 | 100 | 100 | 100 | 100 | 100 | 100 |

Bld-Below detection limit.

The plotted binary relationships of the major oxides (in wt.%) exhibit similar characteristics: on the $SiO_2$ vs. $(Na_2O + K_2O)$ variation diagrams of [28,29]; Figure 8a,b), these granites are alkali feldspar granites except for the El-Missikat sheared granite sample, and two samples from El-Erediya albite granite. The al vs. alk relationship (Figure 8c) exhibits a strong positive correlation, while the normative albite vs. CaO relationship (Figure 8d) shows a strong negative correlation pointing to potassic-sodic alteration. On the $(Al_2O_3 + CaO)/(FeOt + Na_2O + K_2O)$ versus 100 $(MgO + FeOt + TiO_2)/SiO_2$ diagram ([30]; Figure 9a) these granites are chiefly highly fractionated calc-alkaline granite but only two samples possess a calc-alkaline trend. All the samples on the R1-R2 binary diagram of Batchelor and Bowden ([31]; Figure 9b) are plotted within the syn-collisional and late orogenic granites fields. These granites are within plate granites according to the geotectonic classification of [32] (Figure 9c). Such tectonic settings and the associated alteration processes refer to the action of co-magmatic hydrothermal fluids.

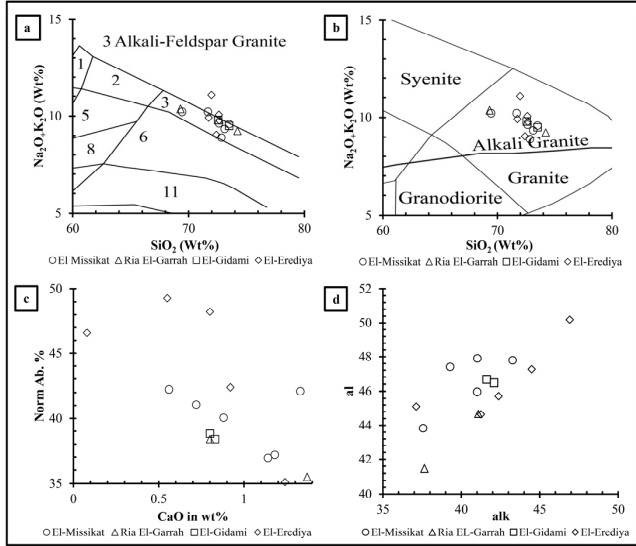

**Figure 8.** (**a**) $SiO_2$ vs. $(Na_2O + K_2O)$ variation diagram for the studied granitic rocks [28], and (**b**) $SiO_2$ vs. $(Na_2O + K_2O)$ variation diagram [29]. (**c**) Variation of albite (norm) vs. CaO binary diagram (**d**) Niggli values al-alk binary diagrams.

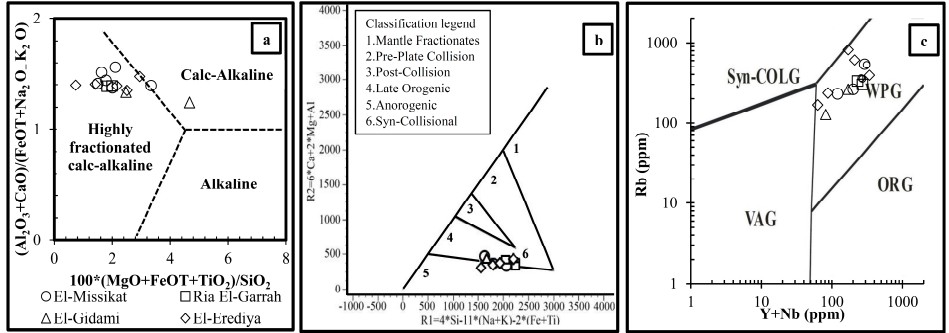

**Figure 9.** (**a**) Plots of the granitic samples on [30] binary diagram, (**b**) R1-R2 binary diagram of [31], (**c**) Rb vs. Nb + Y binary diagram [32], WPG, within-plate granite; ORG, ocean-ridge granite; VAG, volcanic-arc granite; syn-COLG, syn-collision granite.

The chemical analyses representing 29 Y Gr plutons of the central domain (CD) of the eastern desert [16] were used to recognize their geochemical characteristics. The plotted averages of $Fe_2O_3$ vs. MgO and $Fe_2O_3$ vs. CaO diagrams are represented by symbol o in Figure 10; these chemical analyses exhibit a negative correlation. The negative distribution discriminates these averages into three geochemical fields: monzogranite, syenogranite, and alkali feldspar granites. Meanwhile, the K/Na ratio values (Table 2) discriminate the CD Y Gr into these three types.

On plotting the $Fe_2O_3$, MgO, and CaO of the studied samples in (Figure 10), the studied granites are shown to be of the Monzo, syeno, and alkali feldspar types affected by ferrogination.

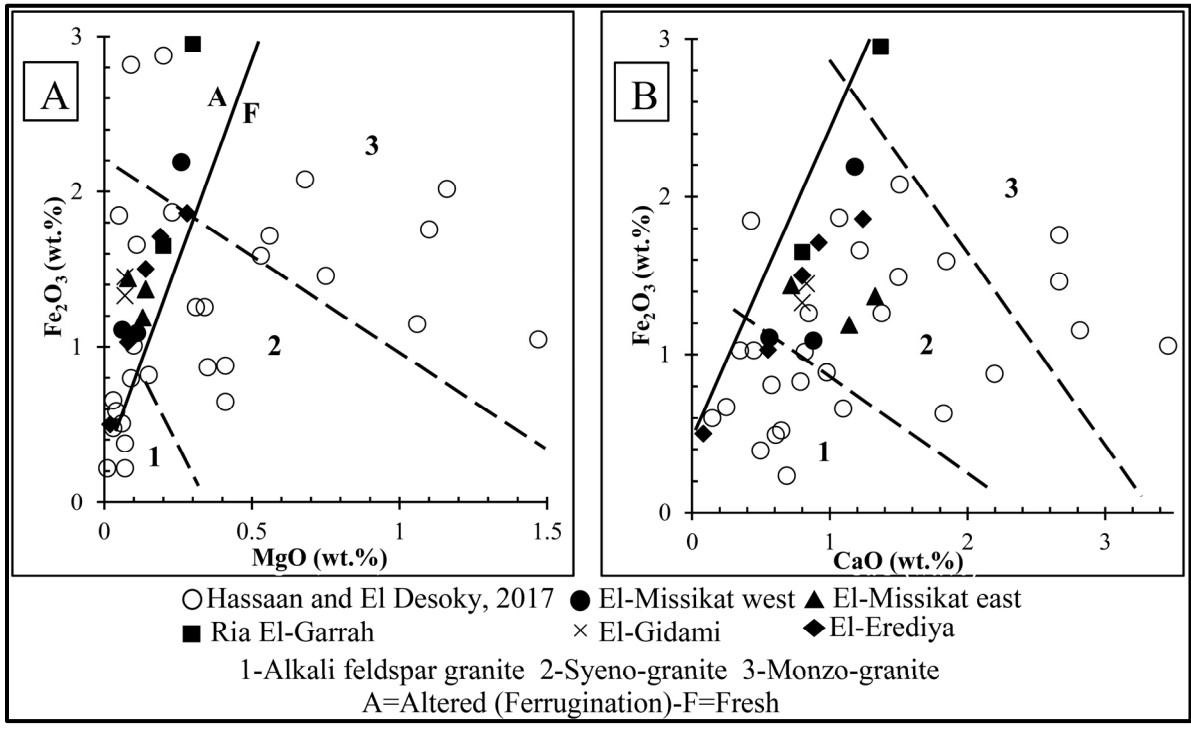

**Figure 10.** Plots of binary relationships (**A**) $Fe_2O_3$ vs. CaO, (**B**) $Fe_2O_3$ vs. MgO of Y Gr in the central-eastern desert (CED) and the studied granites.

**Table 2.** Showing K/Na values of (a) the central-eastern desert and (b) the studied Y. Gr. at El-Missikat-El Erediya.

| Central Eastern Desert Samples | | | The Studied Samples | | |
|---|---|---|---|---|---|
| **Serial No.** | **Area, No. of Samples** | **K/Na** | | | |
| **Monzo-Granite** | | | **Monzo-Granite** | | |
| 27 | Atalla (3) | 0.616 | **Sample No.** | **Area** | **K/Na** |
| 14 | Abu Dabbab (2) | 0.764 | Er5 | El-Erediya | 0.93 |
| 28 | Kib Absi (15) | 0.769 | Er6 | | 0.95 |
| 10 | Igla (2) | 0.827 | | | |
| 22 | Homr akarem (3) | 0.849 | | | |
| 4 | El-Yatima (3) | 0.953 | | | |
| 5 | El-Daghbag (5) | 0.967 | | | |
| 20 | Gabal Ineigi (4) | 0.969 | | | |
| 9 | Gabal El-Ineigi (8) | 0.995 | | | |
| **Syeno-granite** | | | **Syeno-granite** | | |
| 15 | El Dob-Abu Kharif (5) | 1.049 | **Sample No.** | **Area** | **K/Na** |
| 3 | Homrit Waggat (4) | 1.061 | M2 | | 1.176 |
| 8 | Um Gheig (2) | 1.072 | M7 | | 1.31 |
| 26 | Fawakhir (6) | 1.090 | M12 | El-Missikat | 1.238 |
| 13 | Umm Had (2) | 1.101 | M16 | | 1.389 |
| 6 | Um Gheig (5) | 1.152 | M18 | | 1.319 |
| 7 | Um Gheig (4) | 1.204 | M20 | | 1.525 |
| 12 | Umm Had (14) | 1.254 | RG1 | Ria El-Garrah | 1.314 |
| 24 | Um Naggat (15) | 1.267 | G1 | El-Gidami | 1.464 |
| 25 | El Deleihimi (5) | 1.277 | G3 | | 1.362 |
| 18 | Gabal El Sibai-Abu Nagga'a (4) | 1.291 | Er1 | | 1.245 |
| 21 | Homrit waggat (14) | 1.323 | Er3 | El-Erediya | 1.488 |
| 11 | Umm Had (2) | 1.344 | Er4 | | 1.202 |
| 1 | Um Tagher (5) | 1.418 | | | |
| 16 | Gabal El Sibai-Um Lasseifa (6) | 1.418 | | | |
| 2 | Igla Ahmar (8) | 1.442 | | | |
| 17 | Gabal El Sibai-El-Mirifiya (4) | 1.458 | | | |
| 29 | Kadabbora (10) | 1.530 | | | |
| **Alkali feldspar granite** | | | **Alkali feldspar granite** | | |
| 19 | Gabal El Sibai-Um Shaddad (4) | 1.638 | **S. No.** | **Area** | **K/Na** |
| 23 | Um Naggat (6) | 2.564 | RG4 | RiaEl-Garrah | 1.838 |

The chemical components of the representative analyses (except four) of the studied granite plutons are distinguished by $Fe_2O_3$ < 1.5 wt.%, CaO < 1.0 wt.%, and MgO < 0.2 wt.% contents. The four samples possess abnormal higher values of $Fe_2O_3$, CaO, and MgO. The

data reflect the effect of alteration processes, in particular ferrugination. The plotted Na, K, $Na_2O$, and $K_2O$ contents on the diagrams of CTD (Figure 10) show almost scattered distribution, referring to the increased action of the potassic alteration processes. These relationships show that the Ria El-Garrah samples plot within an altered monzo-syenogranite field, while the El-Erediya, El-Gidami, and El-Missikat samples plot within an altered syeno-alkali feldspar granite field. However, both $Fe_2O_3$ vs. MgO and $Fe_2O_3$ vs. CaO relationships reflect the alteration of ferromagnesian and calcic plagioclase minerals. The samples of the four granites plot within the monzogranite to chiefly syeno- alkali feldspar granite fields. The $Na_2O$-$K_2O$ relationship exhibits an obvious degree of sodic- potassic alteration processes on the four granites

The $Fe_2O_3$ vs. CaO, $Fe_2O_3$ vs. MgO, and $Na_2O$-$K_2O$ correlation patterns, which represent the degree of action of magmatic-hydrothermal fluids, characterize this U-F Y Gr geochemical province.

### 3.2.2. Geochemical Characteristics of Trace Elements Distribution

The Clark of concentration, correlation coefficient (r), and zoning coefficient are factors used to measure and reveal their characteristic behavior. These factors delineate zones of anomalous concentrations that may be related to mineral deposits [33].

The calculated CC values of the elements (Table 3) exhibit several characteristics. The siderophile Cr element (CC 4–29) records show extraordinary high contents matching the Y Gr outcrops of the eastern desert [33]. An extraordinary anomalous concentration of Ti (CC = 84) was recorded in two samples of the El-Erediya southeastern perthitic granite mass. These reflect genetic emplacement from deeper levels of the crust: the very low Ni content and extremely abnormal anomalous Ti content in Ria El-Garrah S. No. RG1 refers to its location in contact with metavolcanic and hornblende gabbro outcrops. The content of each of the lithophile Sr and Ba elements is low, recognizing the alkali affinity of the studied granites.

**Table 3.** Clark of concentration of the studied samples and Mo of [8].

| Field No. | Area | Ga | Zr | Ba | Nb | Rb | Sr | Y | Zn | Cr | Cu | Ti | Ni |
|---|---|---|---|---|---|---|---|---|---|---|---|---|---|
| M2 | | 2.9 | 1 | 0.7 | 3.8 | 3.2 | 0.1 | 5.3 | 3.1 | 12.2 | 4 | - | 8.9 |
| M7 | | 2.4 | 1.4 | 0.3 | 2.4 | 1.4 | 1.1 | 1.8 | 5.4 | 3.8 | 3 | - | 6.7 |
| M12 | El Missikat | 2.9 | 1 | 0.1 | 3.3 | 1.5 | 0.3 | 3.3 | 5.1 | 1.7 | 4 | - | 8.9 |
| M16 | | 3.5 | 1.1 | 1.3 | 5.2 | 3.1 | 0.2 | 5 | 3.3 | 1.2 | 2 | - | 13.3 |
| M18 | | 2.9 | 1.3 | 0.5 | 4.8 | 2.1 | 0.2 | 4.3 | 3.1 | 1.9 | 2 | - | 8.9 |
| M20 | | 2.9 | 1.5 | 0.7 | 4.3 | 2.1 | 0.3 | 4.3 | 2.8 | 15.9 | - | - | 6.7 |
| RG1 | Ria El Garrah | 1.3 | 0.6 | 0.04 | 0.8 | 0.8 | 0.1 | 1.6 | 1 | 4.1 | 0.5 | 84.3 | 0.6 |
| RG4 | | 2.4 | 1.6 | 0.3 | 2.9 | 1.5 | 0.8 | 2.8 | 3.1 | 29.3 | - | - | 15.6 |
| G1 | El Gidami | 1.8 | 1 | 0.5 | 3.8 | 1.9 | 0.2 | 3.8 | 9.2 | 3.9 | 3 | - | 11.1 |
| G3 | | 2.9 | 1.3 | 0.4 | 4.8 | 1.8 | 0.2 | 4.3 | 1.8 | 11.2 | 4 | - | 13.3 |
| Er1 | | 6.5 | 3.7 | 0.2 | 3.8 | 4.8 | 0.4 | 2.3 | 17.4 | 9.8 | 4 | - | 11.1 |
| Er3 | | 1.3 | 0.6 | 0.1 | 1.4 | 1.4 | 0.4 | 1.5 | 1.1 | 25.3 | 0.6 | 120.2 | 0.6 |
| Er4 | El Erediya | 1 | 0.4 | 0.1 | 1.2 | 0.98 | 0.8 | 1 | 0.7 | 8.8 | 0.4 | 78.6 | 0.3 |
| Er5 | | 3.5 | 1.6 | 0.1 | 4.8 | 3.6 | 0.2 | 2.8 | 11.5 | 26.8 | 5 | - | 6.7 |
| Er6 | | 4.1 | 1.7 | 0.1 | 5.2 | 2.3 | 0.3 | 5.8 | 6.7 | 4.5 | 5 | 4.3 | 11.1 |
| [8] | | | | | | | | | | | | | |
| S. No. | | 1 | | 2 | | 3 | | 4 | | 5 | | | |
| Mo | | 29.2 | | 34.6 | | 30.8 | | 38.5 | | 26.9 | | | |

The high concentration of Zn (CC 5–17) and Ni (CC 7–16) for most of the samples and moderate Cu concentration (CC 3–5) are recognized in the El-Missikat and El-Erediya samples. Only one sample of each of the Ria El-Garrah and El-Gidami bodies is distinguished by anomalous Ga, Nb, Y, Zn, and Ni contents. These CC values may be attributed to the action of hydrothermal fluids. Gold (0.2 ppm) is recorded by the Nuclear Materials Authority of Egypt (NMA, internal report) in the El-Erediya northwestern part of pegmatitic granite. Moreover, the abnormal, anomalous concentrations of Ga, Zn, Cr, Cu, and Ni are recorded in two samples (No Er1 and No Er5) of these pegmatitic granites. Meanwhile, each of the Y, Nb, Zn, Ga, Cu, and Ni records abnormal, anomalous concentrations in S. No. Er6 of the central porphyritic perthitic granite mass. The elements Ga, Cu, Zn, and Ni, are associate with gold mineralization in the eastern desert [33].

The calculated correlation coefficient (r) matrix (Table 4) exhibited significant confident correlated pairs of elements. Titanium exhibited a negative correlation with each of Ga, Zr, Nb, and Y, referring to its existence as a proper Fe-Ti oxide mineral (ilmenite and rutile) recorded by [34].

**Table 4.** Correlation matrix of the studied samples.

|  | Ga | Zr | Ba | Nb | Rb | Sr | Y | Zn | Cr | Cu | Ti | Ni |
|---|---|---|---|---|---|---|---|---|---|---|---|---|
| **Ga** | 1 | | | | | | | | | | | |
| **Zr** | 0.91 | 1 | | | | | | | | | | |
| **Ba** | 0.14 | −0.04 | 1 | | | | | | | | | |
| **Nb** | 0.64 | 0.41 | 0.55 | 1 | | | | | | | | |
| **Rb** | 0.87 | 0.77 | 0.29 | 0.61 | 1 | | | | | | | |
| **Sr** | −0.19 | 0.05 | −0.25 | −0.44 | −0.32 | 1 | | | | | | |
| **Y** | 0.38 | 0.11 | 0.63 | 0.84 | 0.35 | −0.54 | 1 | | | | | |
| **Zn** | 0.76 | 0.82 | −0.14 | 0.36 | 0.77 | −0.06 | 0 | 1 | | | | |
| **Cr** | −0.08 | 0.09 | −0.23 | −0.14 | 0.09 | 0.14 | −0.26 | 0.02 | 1 | | | |
| **Cu** | 0.59 | 0.4 | −0.08 | 0.59 | 0.55 | −0.28 | 0.43 | 0.6 | −0.23 | 1 | | |
| **Ti** | −0.6 | −0.5 | −0.47 | −0.79 | −0.5 | 0.06 | −0.64 | −0.45 | 0.19 | −0.54 | 1 | |
| **Ni** | 0.54 | 0.48 | 0.48 | 0.73 | 0.43 | −0.08 | 0.64 | 0.32 | −0.05 | 0.39 | −0.82 | 1 |

The correlation of the chalcophile elements of Ga-Zn-Cu-Ni confirms the precipitation of sulfide minerals associated with Au mineralization. The lithophile elements Rb, Sr, Y, Nb, and Zr, are correlated with siderophile and chalcophile elements. This refers to the probable encountering of the presence of Zn, Cu, and Ni within the crystal lattices of altered silicate mineral constituents. These chalcophile elements plotted versus each of $Fe_2O_3$ (Figure 11) and Ga (Figure 12) support the existence of Cu, Zn, and Ni either adsorbed onto hematite and/or as sulfide minerals.

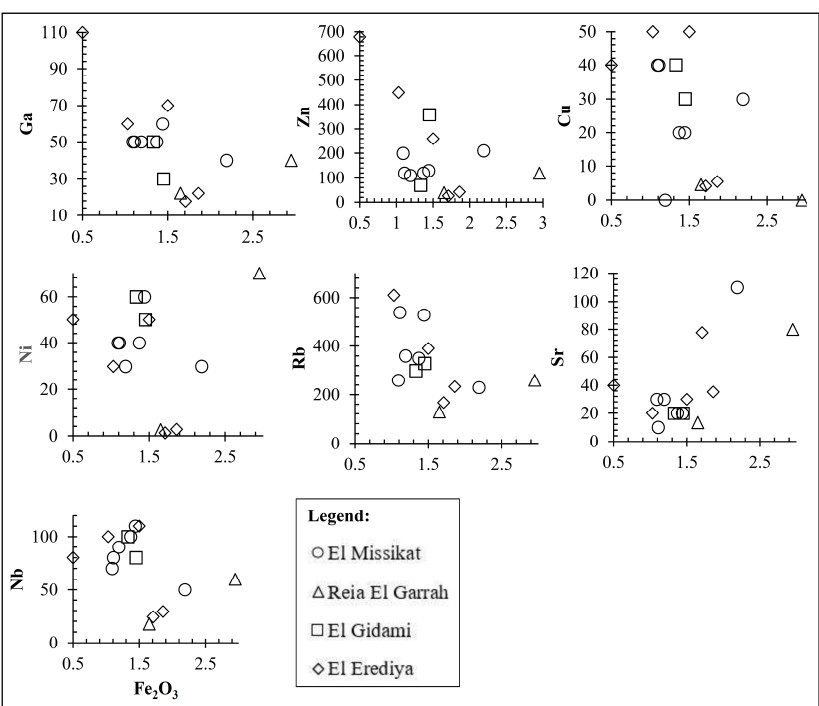

**Figure 11.** The plotted binary relationships of the chalcophile Ga, Zn, Cu, Ni, and lithophile Rb, Sr, Nb elements vs. $Fe_2O_3$ of the studied samples.

In the El-Missikat shear zone, pyrite, chalcopyrite, galena, sphalerite, and pyrrhotite (magnetic iron sulfide) with large gold cubes were found [8,14]. The presence of pyrite and galena in the El-Erediya F-U mineralized zone was recognized [34]. On another side, the positive correlation of Ba, Nb, Rb, Sr, Y, Zr vs. Ga (Figure 12) in some samples favors that these lithophile elements are encountered within the crystal lattice of the silicate minerals. Barite also occurs in the shear zone of El-Missikat granite [14].

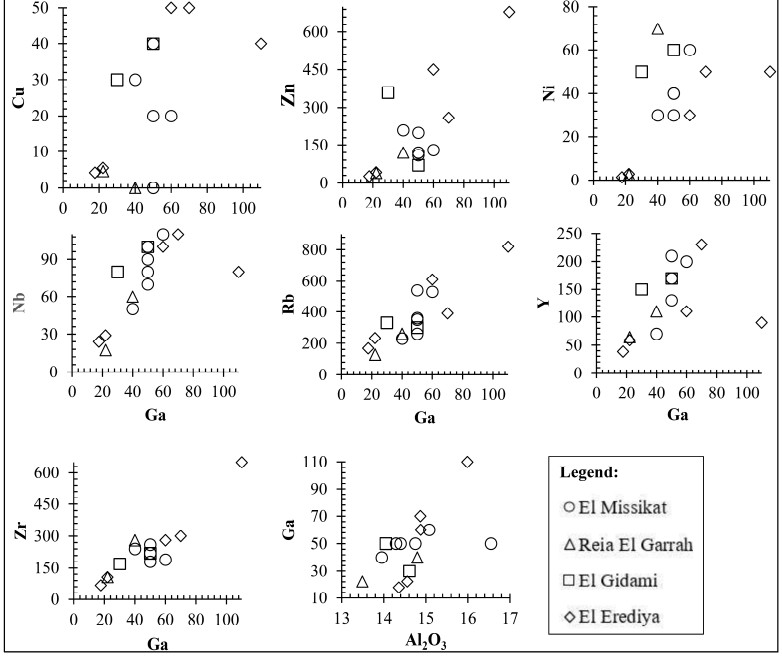

**Figure 12.** Plots of binary relationships of the chalcophile Cu, Zn, Ni and Nb, Rb, Y, Zr elements vs. Ga; Ga vs. $Al_2O_3$ of the studied samples.

The chemical analyses (Table 1) exhibit the Cu (30–40 ppm) in El-Missikat, and (40–50) in El-Erediya. In addition, Ref. [8] abnormal concentrations of molybdenum (35–50 ppm), Pb (85–99 ppm), and U (20–35 ppm) were found in the El-Missikat shear zone and Mo (30–55 ppm), Pb (80–95 ppm), and U (20–30 ppm) in El-Erediya pluton. Furthermore, both Mo and Cu show abnormal CC values (Table 3). These CC values exhibit extraordinary concentrations of Mo and abnormal concentrations of Cu. The plotted correlation diagrams of U, Pb, Cu vs. Mo (Figure 13) show that each of U and Pb records a positive correlation with Mo and a negative correlation of Cu with Mo. This means that molybdenum content increases when Cu decreases. This probably gives evidence of weak Cu-Mo porphyry mineralization in El-Erediya central porphyritic perthite granite. These geochemical features point probably to a hydrothermal Mo porphyry mineralization within El-Erediya central porphyritic perthite granite and polymetallic veins in the El-Missikat ENE to NE highly fractured shear zone. These three modes of occurrence bearing wolframite, molybdenite, scheelite, chalcopyrite, sphalerite, gold and uranophane, and ore minerals [8,14] represent hydrothermal deposits around the magmatic center.

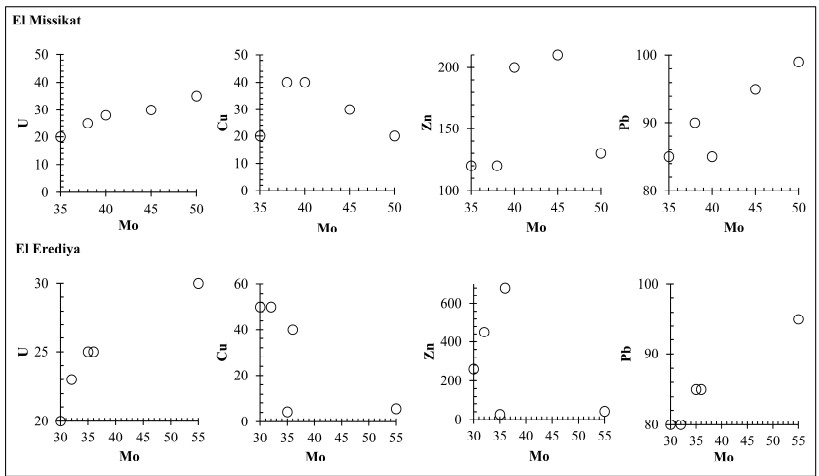

**Figure 13.** Plots of binary relationships of U, Cu, Zn, and Pb vs. Mo in El Missikat and El-Erediya.

## 4. Discussion

### 4.1. Geochemical Pathfinders

The elements and elemental ratios correlation patterns were plotted using the analyses of the collected samples from the studied El-Erediya, Ria El Garrah–El Gidami, and El-Missikat granites, which recorded negative correlation patterns of Rb/Sr vs. K/Na and Rb vs. Sr (Figure 14).

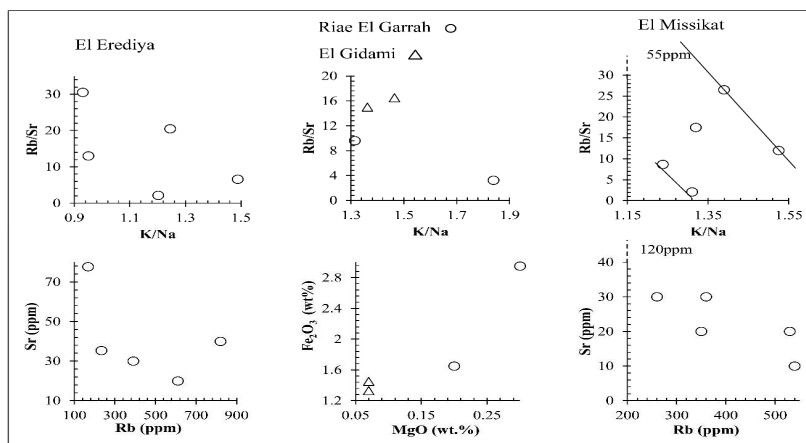

**Figure 14.** Plots of binary relationships of some chemical components of the studied granites.

The plotted patterns of the collected previous data [10,12,13] recognize the positive correlation patterns of K/Na vs. Rb/Sr and Fe$_2$O$_3$ vs. CaO and the negative correlation patterns of Sr vs. Rb, Rb/Sr vs. K/Na, and K vs. Zr (Figure 15).

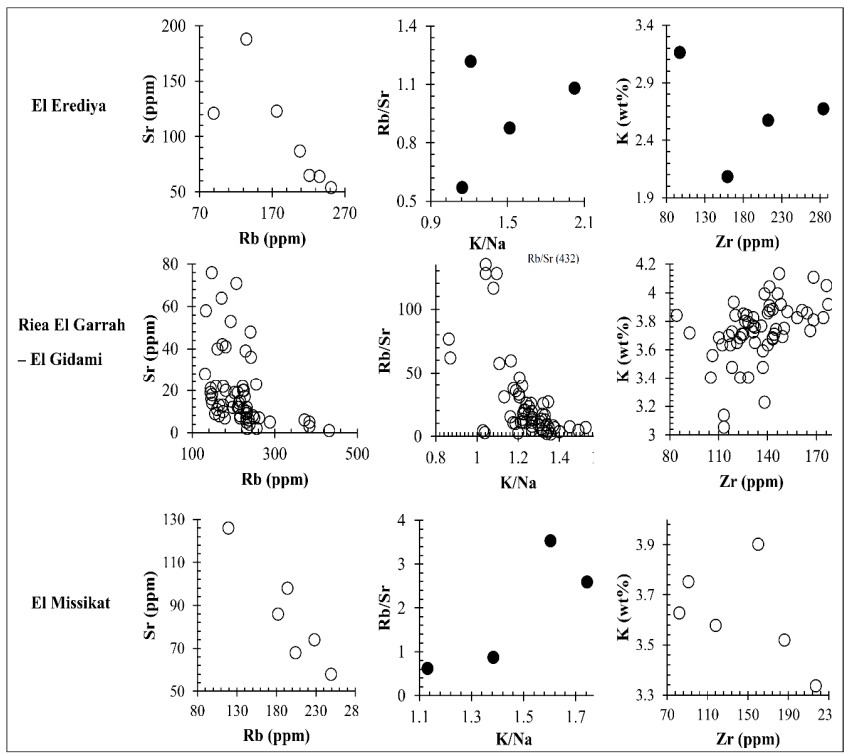

**Figure 15.** Plots of binary relationships from previous chemical analyses, El-Erediya granites [12]; Ria El–Garrah and Gidami granites [10] and El-Missikat granites [12,13].

Only the Sr vs. Rb and Rb/Sr vs. K/Na negative correlation patterns of both data show similar negative patterns with eU vs. Fe$_2$O$_3$, eU vs. K, and cU vs. Al$_2$O$_3$ patterns (eU = equivalent U; cU = chemical U; Figure 16). Similarly, Fe$_2$O$_3$ vs. MgO, eU vs. Al$_2$O$_3$, and eU vs. Fe$_2$O$_3$ exhibit negative correlation patterns. In consequence, the Sr vs. Rb, Rb/Sr vs. K/Na, and Fe$_2$O$_3$ vs. MgO negative correlation patterns are considered geochemical pathfinders to explore for U deposits in the eastern desert Y Gr bodies.

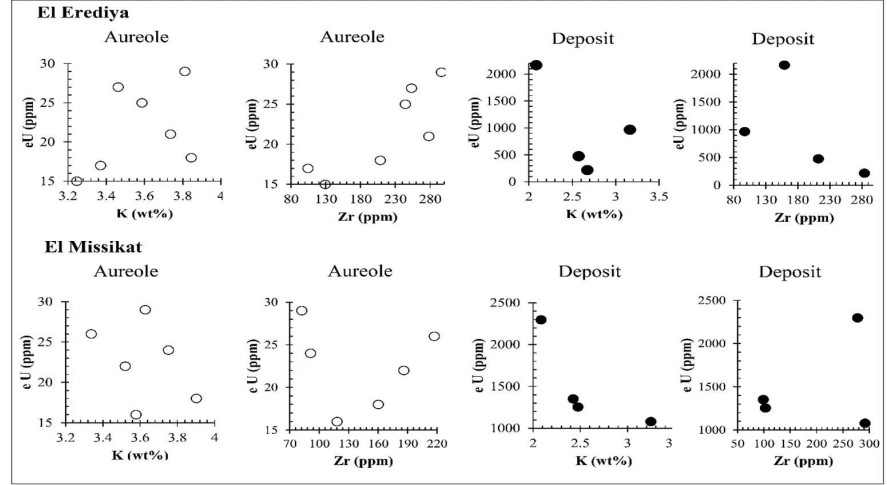

**Figure 16.** Plots of binary relationships of eU vs. K and Zr in El Erediya and El Missikat aureole and deposit [12].

Chemical analyses of the four studied Y Gr bodies in the previous studies show samples containing eU < 30 ppm and eU > 30 ppm up to 2900 ppm in only El-Erediya, and the El-Missikat samples are classified into uraniferous and fertile zones [35]. From this account, the contents of cU, eU, eTh (c = chemical; e = equivalent), major oxides, and some trace elements given in these previous studies are used to reveal the grade of radiogenic decay of proper U minerals. The calculated eU/cU values of five of the six samples range from 2.1–2.7. The sample containing only 4 ppm cU exhibits an abnormal ratio due to a high rate of radiogenic decay. This range of ratio values can be applied for confident radiometric analysis; hence, the plotted correlation diagrams using eU values are considered significant.

The present study considers the fertile and uraniferous zones of each of the El-Erediya and El-Missikat deposits and their aureole. The plotted Sr vs. Rb, K vs. Zr, eU vs. K, and eU vs. Zr (Figures 15 and 16) represent variable patterns. The positive correlation between K/Na vs. Rb/Sr, Sr vs. Rb, and K vs. Zr is similar to eU vs. K, and eU vs. Zr discriminates two fields: the deposit (eU > 30 ppm) and the aureole (eU < 30 ppm). The plotting of eU vs. K and eU vs. Zr using a more detailed interval (Figure 16) confirmed the relation of K/Na vs. Rb/Sr, Sr vs. Rb, and K vs. Zr. In consequence, these positive correlation patterns could be pathfinders for delineating other hydrothermal Y Gr plutons bearing uranium deposits in the eastern desert of Egypt.

The values of the Zr/Sr ratio (from 0.48 to 5.48), in addition to the presence of fluorite (pockets, patches and veins), reflect that El-Erediya is U-rich deposit and the Zr/Sr ratio (from 0.72 to 2.93) in El-Missikat plutons' U-poor zone [34]. The calculated values of the Zr/Sr ratio of the analyzed samples of both plutons range from 1.65-up to 5.48 and up to 2.93, respectively. These Zr/Sr ratio values recognized the U-rich and U-poor deposits of El-Erediya and El-Missikat, respectively, from uriniferous rocks. The two plutons that exhibit Zr/Sr > 1, followed by [36], are advanced magmatic differentiated granites. In consequence, Zr/Sr values plus the presence of fluorites can be considered a geochemical pathfinder factor for advanced magmatic differentiated granites hosting uranium deposits in the eastern desert.

*4.2. Uranium Deposits*

The Clark of Concentration (CC), correlation coefficient (r), and zoning coefficient (γ) are used to recognize the sequence of deposition of elements of mineral deposits. These factors are reliable for revealing the constituent elements, proper chief element and its associated elements, and vertical site of the center of enrichment of its chief metal from the surface of the erosional crust of a mineral deposit and its lateral extension. These three factors are applied to study El-Erediya and El-Missikat uranium deposits.

The calculated Clark of Concentration (CC) of each element divided by its average content of Y Gr is summed up for each element. The arrangement of CC values is expressed in the arbitral form: $\frac{A}{B}$ C (D) where A is the symbol of the chief element having ΣCC values > 0.5 ΣCC values of all elements, B is the principal associated element having ΣCC values < 0.5 > 0.25 ΣCC values, C is the associated elements having ΣCC values < 0.25 > 0.05 ΣCC values, and D is the elements having CC value < 1. The recorded elements occupying the A, B, and C sites are the associated elements of the U mineral deposit.

The calculated CC values produced the geochemical spectrum $\frac{U}{Th}$ Cu, Pb (Zn) for both U deposits. This geochemical spectrum recognizes U as the chief ore element, Th as the principal associated element, and Cu and Pb as the associated elements, while Zn exhibits normal dispersed content. The elements Th, Cu, and Pb, represent the geochemical association of both uranium deposits. Additionally, the existence of anomalous concentrations of Mo (30–55 ppm), Pb (80–95 ppm), Cu (40–50 ppm), and U (20–30 ppm) (associated with W, Sn, and F) is mentioned by [8] in the greisen lenses (after granite) of the El-Missikat shear zone. Their CC value ranges are Mo 27–39, Pb 7–8, U 5–7, and Cu 3–6. These greisen lenses exist in association with polymetallic quartz veins bearing gold cubes, wolframite, chalcopyrite, galena, and cassiterite [14]. Anomalous concentrations of Mo 35–55 ppm,

Pb 85–99 ppm, Cu 30–40 ppm, and U 20–35 ppm are recognized around the El-Erediya uranium zone. Their CC value ranges are Mo 27 to 43, Pb 5 to 8, U 5 to 9, and Cu 4 to 5. These CC values exhibit the extraordinary concentration of Mo and abnormal concentration of Pb, U, and Cu.

The correlation coefficient (r) shows the positive correlation patterns of each of U and Pb, and the negative correlation pattern of Cu with Mo (Figure 13) points to an increase of Mo, U, and Pb contents with depth while the Cu content decreases. This may give evidence for the probable presence of Mo porphyry deposits at deeper levels. Molybdenum porphyry deposit probably exists with pervasive hydrothermal alteration in and immediately around the stock of the porphyritic perthite granite. This points to the fact that porphyritic perthite granite is intruded following [27] to shallow levels in the crust.

The zoning coefficient ($\gamma$) is a factor that can be calculated to express the zonal distribution of ore element constituents of a mineral deposit [33]. The zoning coefficient ($\gamma$) manifests the change from the center or periphery of the ore body in $C_i$, Cci, or productivity $M_i$ of the chief and associated elements relative to those of the inert or chief ore element (CC $_{inert}$), (M $_{inert}$) at different vertical distances for each drilled hole or lateral distances between drilled holes, respectively. This factor is calculated using the following formula:

$$\text{Zoning Coefficient } \gamma = \frac{Ci}{Ci \; inert} \text{ or } \frac{Mi \; 1,2,3}{Mi} = f(X) \text{ or CCi 1, 2, 3/CC inert}$$

where $Ci$—the content of the element, CCi—Clark of Concentration, $Mi$ (linear productivity of the chief or inert element) = $\Sigma(Ci - \Delta X) = Mi$ $_{1,2,3}$, $Mi$, and $\Delta X$ = half distance between preceding and following samples. Calculated $\gamma$ values for each element versus vertical sites of the samples are then plotted, drawing the averaging regression for each element and arranging of the averaging regression lines of the element to produce the vertical zoning sequence of deposition of elements for each drill-hole from the deepest level upwards. The reached zoning sequences of deposition of elements almost followed, with some exceptional deviation, the general sequence of deposition of elements from the deepest level upward given in [33].

Zoning coefficient ($\gamma$) values are calculated using the chemical data of five drilled boreholes [34]; two penetrating the El-Missikat shear zone and three penetrating the El-Erydiya porphyritic perthite granite body. The ($\gamma$) values are vertically plotted for each borehole with the depth and the lateral sites for the boreholes of each body to reveal whether the vertical and lateral geochemical sequence of zoning of deposition controls the deposition of these metals. The produced zoning sequences will express a better understanding of the genesis, structure of the ore-field, level of the erosional crust, and its extension for both the El-Missikat and El-Erediya Y Gr bodies.

### 4.2.1. El-Missikat

The plotted zoning coefficient (v) values of U/Pb, Cu/Pb, and Zn/Pb versus the depth of the raised samples from two drilled boreholes penetrating El-Missikat and four penetrating El-Erediya are given in Figures 17a,b and 18a–d.

The arrangement of the obtained to derived averaging regression lines of the plotted values for each of the El-Missikat boreholes (Ms2 and Ms3) suggests the presence of vertical zoning sequences of deposition from the deeper levels up to the level 60 m and oxidizing zone from level 60 m up to wadi level (660 m a.s.l.) as follows: Borehole (Ms 2), from level 130 m up to level 60 m, shows Cu, Zn, Th, U, and Pb deposition sequences (Figure 17a). Borehole (Ms 3), from level 150 m up to level 60 m, shows Cu, Zn, Th, U, and Pb (Figure 17b). Boreholes (Ms 2 and Ms 3) from level 60 m up to wadi level are in the oxidized zone.

Borehole Ms 2 comprise the normal vertical zoning sequence of deposition of these elements, representing the simultaneous deposition of these elements with the crystallized magma that formed uraniferous granite. Meanwhile, Borehole Ms 3 shows the additive deposition of U at the level deeper than 60 m, reflecting the role of released hydrothermal fluids from the core of the crystallized magma forming the U-rich zone. The plotted

diagrams of both boreholes point to the presence of an oxidation zone deeper than level 60 m, a transition zone probably reducing around this depth, and the upper 60 m, an oxidizing zone where U and Cu are combined, forming torbernite $Cu(UO_2)_2(PO_4)_2.12H_2O$. The arranged averaging regression lines of lateral zoning sequence of the two boreholes penetrating El-Missikat (Figure 19A) is U, Th, Cu-Zn, and Pb. In consequence, the El-Missikat plotted lateral zoning of the two boreholes points to the fact that borehole Ms-3 is penetrating a U-rich zone (possibly U-weak deposit) while Ms-2 penetrates U, a poor zone.

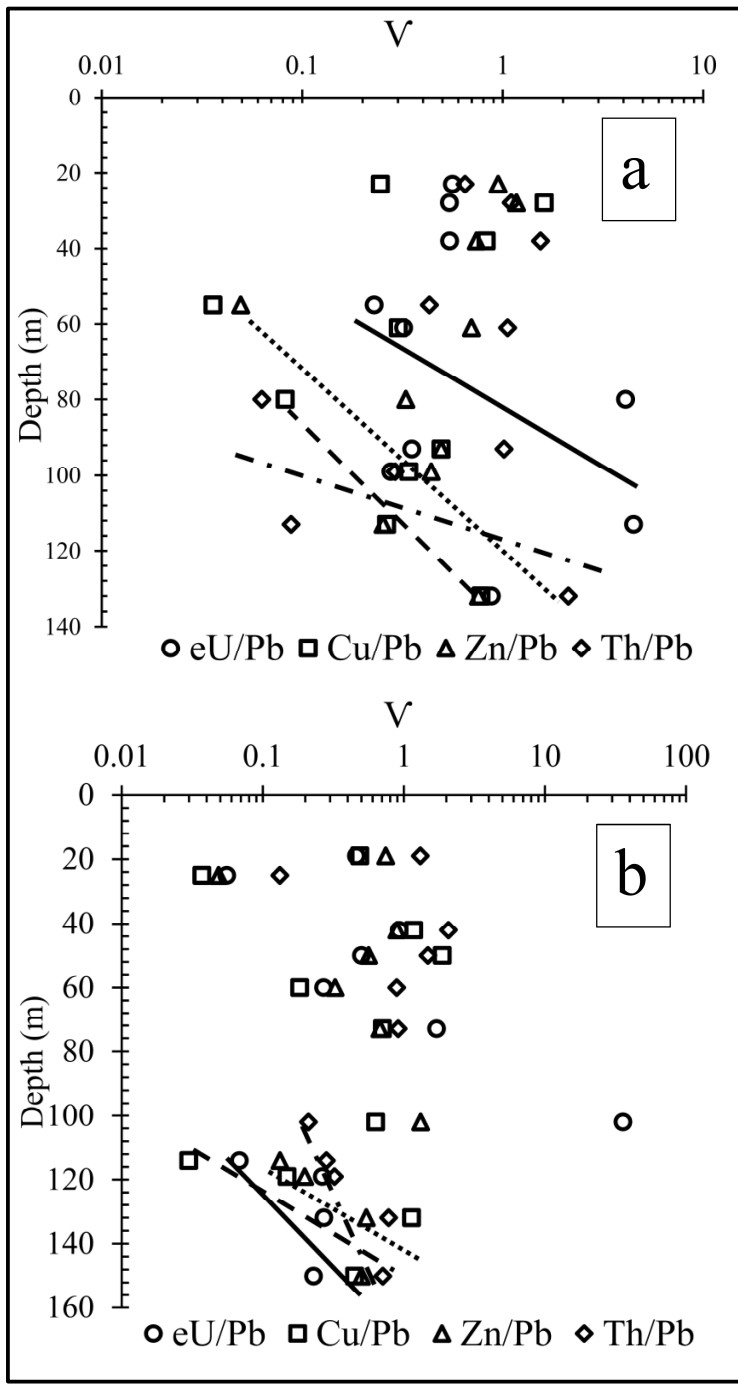

**Figure 17.** Vertical geochemical zoning of eU, Cu, Zn, Pb, and Th of El-Missikat boreholes (**a**) MS2, (**b**) MS3.

### 4.2.2. El-Erediya

The arrangement of the obtained averaging regression lines of each of the four El-Erediya boreholes (Er-1; Er-7M-60; Er-8M90; Er-4M-30) records the following vertical zoning sequences of deposition of these elements from the deepest levels up to the wadi level. Er-1is Cu, U, Zn, Th, Pb, Er-7-M60 is Cu -Zn, Th, Pb, U, and Er-8M-90 is Cu, Zn, U, Th, Pb. Er-4M-30 represents a redistribution of the elements in a secondary oxidation environment (Figure 18a–d).

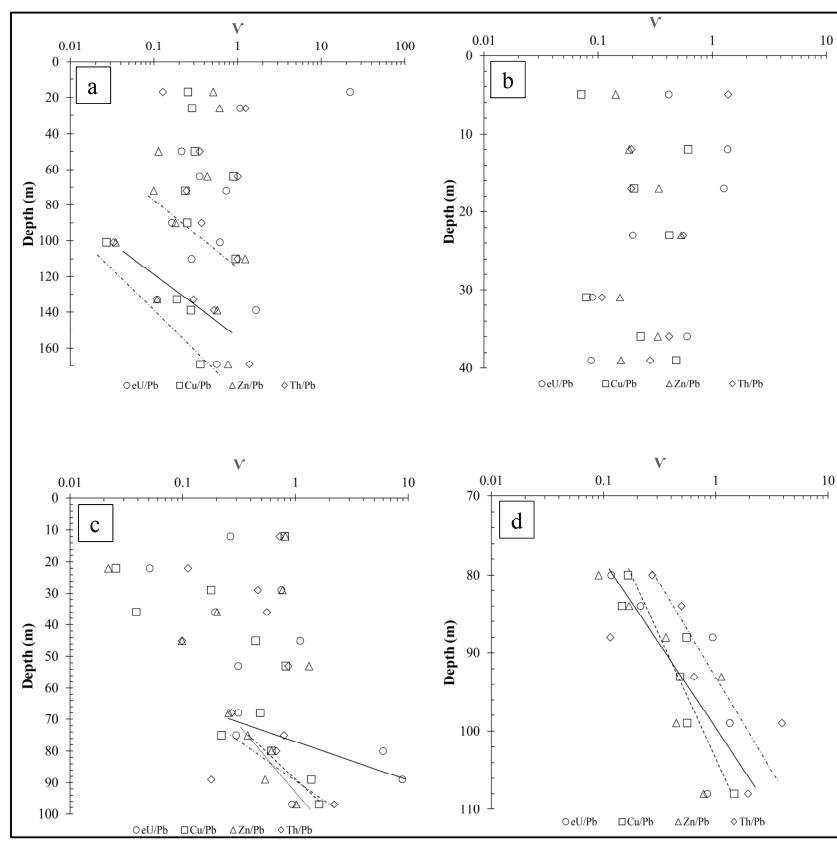

**Figure 18.** Vertical geochemical zoning of eU, Cu, Zn, Pb, and Th of El-Erediya boreholes, (**a**) Er1, (**b**) Er-4M-30, (**c**) Er-7M-60, and (**d**) Er-8M-90.

The normal vertical zoning sequence of deposition for borehole Er-1 points to the simultaneous deposition of U, Th, and Pb with the crystallized magma that penetrates uraniferous granite. In this respect, the averaging regression line of Zn/Pb reflects the enrichment of Zn by the leaching of its dispersed average content in the host granite by convicting heated fluids. Meanwhile, the averaging progressive lines of the Cu/Pb and Zn/Pb of the raised samples from the zone above 60 m express that they are formed by convicting oxidized heated external fluids. The plotted zoning coefficient ($\gamma_{Th/Pb}$, $\gamma_{U/Pb}$, $\gamma_{Cu/Pb}$, $\gamma_{Zn/Pb}$) versus the sites of the four boreholes drilled at El-Erediya shown in Figure 19B records that the lateral sequence of zoning is Cu, Zn, Th, U, and Pb. On the other hand, the El-Erediya plotted lateral zoning of deposition (Figure 19B) points to the fact that borehole Er-8M-90 is penetrating the upper part of the center of the U-rich fertile granite and may be a deposit, whereas boreholes Er-1and Er-7M-60 are penetrating the periphery of the U-rich fertile zone and could be the aureole.

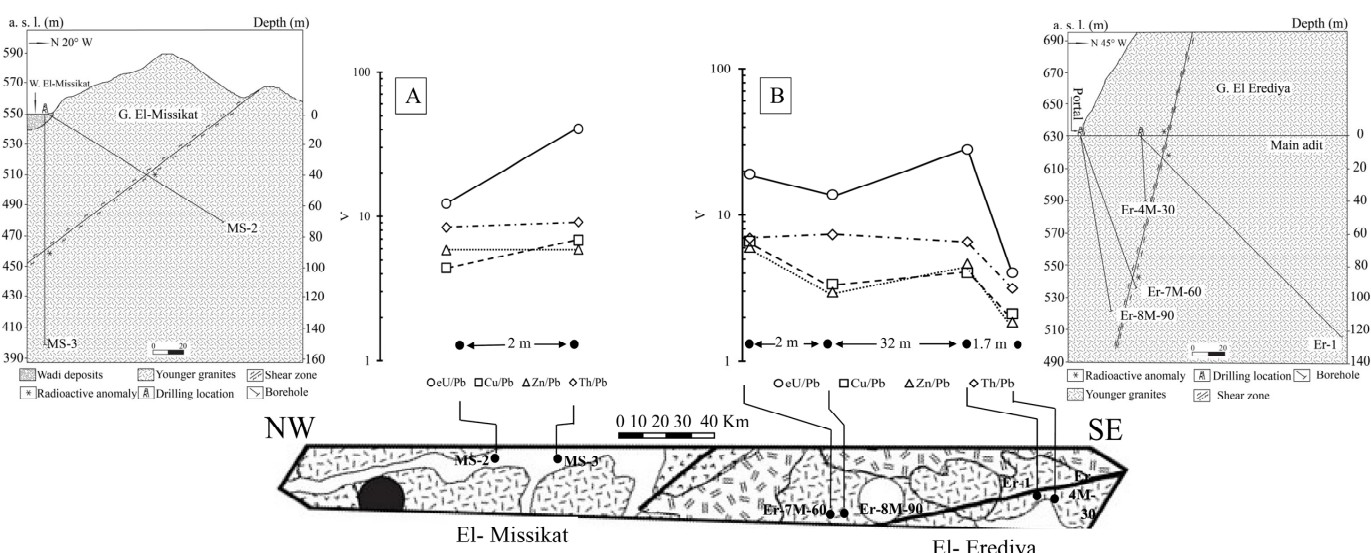

**Figure 19.** Lateral geochemical zoning of eU, Cu, Zn, and Pb for (**A**) El-Missikat and (**B**) El Erediya boreholes.

### 4.3. Genesis of the Mineral Deposits

The modes of occurrence of the U and F deposits, Mo porphyry, Au-Ulfide, and W-Mo greisen, polymetallic vein deposits are associated with the formation of the porphyritic perthite granite central mass of El-Erediya pluton.

Borehole Er-8M-90 geochemically delineated the core of the El-Erediya uranium deposit at deeper subsurface levels at the contact between porphyritic perthite granite and the perthite granite. This borehole recognizes that the deposit is formed by hydrothermal-heated fluid in the core of the hydrothermal system. Boreholes Er-1 and Er-7M-60 penetrate their associated aureole. There, Mo-porphyry mineralization at deeper levels is also probably supported by the extraordinary CC values and positive correlation patterns of Mo vs. Pb and U, and negative for Cu vs. Mo (Figure 12). Moreover, borehole Ms-3 penetrates U-weak deposits at its southeastern subsurface levels. Ria El-Garrah and El-Gidami-uraniferous younger granite plutons are hosted by uranophane, beta-uranophane, and kasolite–fluorite mineralized zones bearing pyrite, galena, chalcopyrite, and sphalerite. El-Missikat shear zones are hosted by greisen lenses, polymetallic veins, and stockwork that bear sulfide-gold particles and Mo, W, Pb, As, and Cu sulfide mineral constituents. Moreover, colorless-purple to deep purple fluorite pockets are excited. The four Y Gr outcrops, El-Erediya, El-Gidami, Ria El Garra, and El-Missikat, represent the geochemical province.

The recognized alteration processes are argillic, phyllic, propylitic chlorite-sericite, and sodic–potassic types. The intensive alteration processes began their development with the emplacement of the El-Erediya porphyritic perthitic granitic mass. The decrease of temperature during migrated magmatic-hydrothermal fluids extended northwestwards until El-Missikat (about 40 km) produced such alteration and associated deposits. The probable precipitation of sulfide mineral constituents of Ga, Cu, Zn, and Ni from magmatic-hydrothermal fluids around the magmatic center is supported by Cu, Zn, and Ni versus each of Ga and $Fe_2O_3$ positive correlation of some of the samples to be considered a geochemical association of the uranium-sulfide gold deposits of such Y Gr bodies.

The characteristics of each of the geochemical distribution, elemental composition, geochemical association, and vertical–lateral sequences of the geochemical zoning of the deposition of U, Th, Cu, Zn, and Pb are related to the oxide and sulfide ore mineral constituents in the studied deposits of granites viz; (a) uranophane, beta-uranophane, kasolite in the upper 80 m levels, uranothorite, and cassiterite at levels deeper than 150 m in only El-Erediya, (b) the secondary fine quartz grains produced by loss zircon of $SiO_2$ during the addition of U and Th to possess its darker color, (c) the calcite produced due to complete replacement of Pb to Ca in uranophane and beta-uranophane that formed

kasolite and, (d) recorded uranothorite and cassiterite at deeper depths (>150 m) pointing to deposition by hydrothermal fluids in oxidation zone followed upward by reduction zone to precipitate the pyrite, galena, and associated at Ria El-Garra and El Gidami with chalcopyrite and sphalerite.

The El-Erediya vertical zoning sequences of deposition of these elements of the three boreholes from deeper levels up to the wadi level represent a normal co-magmatic sequence of deposition of these elements during the formation of the penetrated granite. Meanwhile, the averaging progressive lines of Cu/Pb and Zn/Pb in the upper 60 m levels express deposition by rushed heated ocean water and minerals towards the upper continental rocks.

The fluorites are colorless, purple to deep purple, filling the interstitial spaces of smoky to black jasperoid quartz veins. Fluorite occasionally enclosed in muscovite flakes (after biotite.) reflects the leaching of biotite $\{K (Mg, Fe)_3 (AlSi_3O_{10}) (F, OH)_2\}$ by convicting fluids. The liberated fluorine leaching from biotite is combined with liberated calcium due to Pb replacement in uranophane and beta-uranophane that formed the coarse-grained fluorite in between mica flakes and the kasolite [37].

During EAO, the change from compression to an extension regime was associated with the U-F-Au deposits in the studied district. During subduction, granitoid magmatic-hydrothermal fluids in the core of the hydrothermal system precipitated the main U and Mo porphyry deposits at El-Erediya and convected the heated water of the Mozambique Ocean and rushed minerals deposited U poor deposits at Ria El-Garrah, El-Gidami, and El-Missikat.

*4.4. Correlation of Western Arch Mineralization*

Uranium mineralization and polymetallic veins (Mo, Th, Nb, Ta, Ba) are hosted in multistage deformation pegmatitic granites in Gebel El-Urf ([38]; Figure 20). Gebel El-Dob Monzo-to syeno-Y Gr bears wolframite while wolframite and molybdenite are found in Gebel Abu-Kharif alkali feldspar Y Gr as a stockwork-like shape [33]. Gebel El-Dob Monzo-to syeno-Y Gr is influenced by the same alterations process of the studied granites.

Moreover, the Gebel Gattar and Wadi El-Faliq (northern and southern) Y Gr outcrops bearing U, F, Mo, Ag, and Au filling vein-type deposits structurally trending ENE, NS, and NW linked with tectonic deformations [33,39]. The Gebel Gatter alkali feldspar Y Gr of 604.8 ± 3.3 Ma age is emplaced along a major weak structural zone trending NNW–SSE ([40]; Figure 20). Gebel Gatter Y Gr is subjected to albitization, silicification, epidotization, chloritization, muscovitization, kaolinization, ferrugination, carbonization, and fluoritization. In addition to fluorite, molybdenite, chalcopyrite, sphalerite, wolframite, and goethite after pyrite, bismuthinite, galena, and gold are recorded (Mo 34–66 ppm, Sn 338–548 ppm, W 34–65 ppm, Pb 10–17 ppm, Co 16–36 ppm, Ni 10–31 ppm, Sb 136–220 ppm, Ag 3–11 ppm and Au 0.45–1.03 ppm), pointing to the presence of deposits of Sn-W-Mo as well as Ag- Au beside the F-U deposits [33]. However, at an old excavated tunnel at the wadi level (662 m a.s.l., between Long. 33°16' and Lat. 27°6'), gold increases downwards to reach 3 ppm in a tunnel sample, whereas Mo-Ag contents increase upward to the location of the molybdenite deposit forming zoning sequence of deposition. Additionally, uranium occurrence exists at the site of the excavated tunnel [39].

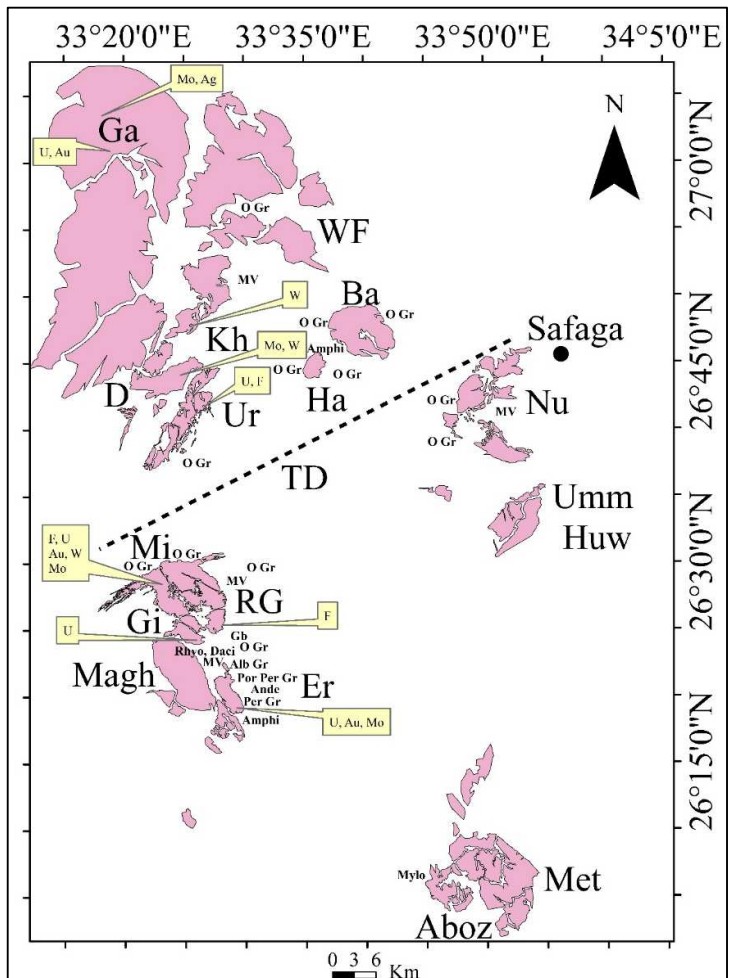

**Figure 20.** Metal associations of granitoid-related ores in the granitic arches: Western arch El-Erediya (Er), Maghrabiya (Magh), El-Gidami (Gi), Ria El-Garrah (RG), El-Missikat (Mi), El-Urf (Ur), El-Dob (D), Abu-Kharif (Kh), Gattar (Ga)), and Eastern arch Umm-Huwitat (Umm Huw), Nuqara (Nu), Abu-Hawies (Ha), Ras-Baroud (Ba), Wadi El-Faliq (WF). Abo Ziran (Aboz), Meatique (Met).

These results give points of evidence that these porphyry- greisen- polymetallic vein deposits, hosted by the WA YGr outcrops of El-Erediya, Ria El-Garra, El-Gidami, El-Missikat, G. El-Urf, G. El-Dob G. Abu-Kharif, and G. Gattar, belong to the hydrothermal ore deposits formed around granite center. In El-Missikat the greisen and the related W-Sn, Mo, U, and F deposits exist at the top of the intrusion and in sheeted quartz veins in and adjacent to altered granite similar to that mentioned by [27]. For the El-Erediya pluton, the increase of Mo, U, Pb, and decreased Cu contents with depth are evidence for probable extended Mo porphyry deposits at deeper levels that are supported by their CC value. Such Mo porphyry deposit probably exists with pervasive hydrothermal alteration in and immediately around the stock of porphyritic granite that are intruded to shallow levels in the crust [27].

In accordance, the Y Gr outcrops of the WA host northwestwards; El-Erediya U-Mo-Au, Ria-El-Garrah-El-Gidami U-F, El-Missikat U-F-Au-Mo-W, El-Urf pegmatites U-F, El-Dob Mo-W, Abu-Kharief W, and Gattar U- Mo-Au-Ag deposits (Figure 20). These Y Gr plutons of the WA represent a geochemical belt that is promising for undertaking such detailed studies supported by drilling programs. Since Wadi El Faliq's northern and southern Y Gr outcrops bear U-F-Au mineralization, the Y Gr outcrops of the EA are also recommended for undertaking such an exploration program for EA granites.

## 5. Conclusions

The Y Gr ellipsoid constituting both WA and EA of 150 km long and 50 km in width represents a subducted continental margin (cordillera). The associated hydrothermal core fluids of Cordillera magmatism and the heated, rushed Mozambique Ocean water and rock minerals filtered by upper continental rocks formed, respectively, deposits in reducing deeper zones and oxidizing at upper levels. The WA Y Gr outcrops recognize an enormous geochemical belt. The studied granites are geochemically altered Monzo-, syeno-, and alkali feldspar, have an advanced differentiated calc-alkaline affinity, and are formed in the active continental margin. They are subjected to argillic, propyllitic, chlorite-sericite, phyllic, sodic-calcic, and potassic alteration processes that represent a proper geochemical province of U-F-Au-Mo-W hydrothermal mineral deposits. Mo porphyry, U-F-Au-W-Sn greisen zones, and F-U-Au-W-Mo polymetallic vein and stockwork veins are formed in metallogenic provinces of El-Erediya, El-Gidami, Ria El-Garrah, and El-Missikat. Two fluids are involved in the hydrothermal system, including magmatic-hydrothermal fluids in the core of crystalized magma and convicting heated oceanic water in the outer alteration zones. The vertical and lateral geochemical zoning sequences of deposition of the metallic elements discriminate the uranium deposits from their U-aureoles, differentiate U-rich from U-poor zones, and define the center of both El- Missikat and El-Erediya deposits. The $Fe_2O_3$ vs. MgO positive correlation patterns, and the negative ones of Rb vs. Sr, also, the distribution patterns of Cu, Zn, and Pb with Ga, as well as the decrease of Cu content with depth where Mo and U contents increase, are proved by the presence of sulfide minerals. These relationsare exploration pathfinders for the delineation of other similar geochemical provinces in the eastern desert.

**Author Contributions:** Conceptualization, M.M.H., A.E.K. and I.M.E.-N.; methodology, A.E.K. and I.M.E.-N.; software, T.M.S.; validation, M.I.S. and M.Y.H.; formal analysis, M.M.H. and S.A.O.; investigation, S.A.O., A.E.K. and T.M.S.; resources; I.M.E.-N. and T.M.S.; data curation, M.M.H. and I.M.E.-N.; writing—original draft preparation, M.M.H., S.A.O., A.E.K. and I.M.E.-N.; writing— review and editing, M.M.H., S.A.O., A.E.K., I.M.E.-N. and M.Y.H.; visualization, A.E.K. and T.M.S.; supervision, M.M.H. and M.I.S.; project administration, S.A.O. and I.M.E.-N. All authors have read and agreed to the published version of the manuscript.

**Funding:** Not applicable.

**Data Availability Statement:** Not applicable.

**Conflicts of Interest:** The authors declare no conflict of interest.

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
