# Peer review of "Prognostic Exploration of U-F-Au-Mo-W Younger Granites for Geochemical Pathfinders, Genetic Affiliations, and Tectonic Setting in El-Erediya-El-Missikat Province, Eastern Desert, Egypt"

_minerals, doi:10.3390/min12050518_

Round 1

Reviewer 1 Report

Very small corrections are needed

(see the text)

Author Response

Dear Reviewer,

Please find attached the submission of the carefully revised version of the manuscript in Ref., following the minor comments and modification of the Reviewer.

Below is a detailed list of the changes made in response to the Reviewer’s minor comments (in italics), which outlines every change made a point by point. The changes are marked in the manuscript text (yellow highlighted).

Please check the manuscript for all comments.

Response: All corrections are done in the manuscript according to the reviewer’s comments are detected in the attached file.

We thank the Reviewer a lot for the useful and valuable comments that have helped improve the manuscript.

Hoping that all the careful review is sufficient for the direct acceptance of the manuscript, thank you for your time and consideration.

Best wishes,

Mohamed. Y. M. Hanfi

on behalf of all co-authors

Reviewer 2 Report

Please see comments enclosed in the attched file.

Author Response

(The authors gave the same response as above.)

Round 2

Reviewer 2 Report

It is a credit to the authors that they have worked very hard to implement changes to make this a more readable article. It must have been a daunting and difficult task. While not perfect the manuscript is now more readable. I have highlighted in yellow a copy of the manuscript to show where some other changes can be made. I have removed capitalisation from eastern and western deserts- you may wish to keep these though which is fine if they are consistent. I attach some small changes and I have highlighted in the text where these are. I am happy to accept the manuscript after the changes have been done. Very well done to the authors who have worked so hard to put this work into English which can be very difficult. I congratulate you.  

L 53 change ended to concluded by

Finally the post-collisional within-plate A-type granite (rie-64 beckite granite) generated during orogenic collapse (not a sentence needs a verb)

For to of line 82 remove discriminating

Line 92 insert which

L95 close bracket

L100 delete applied

L114 remove on other hand

L122 remove occurred

L125 insert are

L126 matches

L168 insert are

L170 insert and

L213 separates

L228 hosts

L249ref after technique

L256 colon not full stop

L274 insert the

L279 remove meanwhile

L317 insert are shown to be

L319 insert the

Consistency with capital letters or not

L329 inset a or the

L337 del are suggested to

L344 records to shows

L346 space after CC and insert the

L350 comma after low

L352 Clarke

L354 recognised to seen

L355 insert the

L361 line missing

L373 encountering to presence

L380 insert the

L389 start with The

L429 del given and mentioned

L432 on to from

L458 Clarke

L493 capital c for consistency

L526 obtained to derived

L575 del whereas and capitalise Boreholes

L596 del obtained

L645 besides to Additionally

L657 insert the

L671 decapitalise associated

Author Response

Please find attached the submission of the carefully revised version of the manuscript in Ref., following the minor comments and modification of the Reviewer.

The changes are marked in the manuscript text (tracks).

Please check the manuscript for all comments.

Response: All corrections are done in the manuscript according to the reviewer’s comments are detected in the attached file.

We thank the Reviewer a lot for the useful and valuable comments that have helped improve the manuscript.

Hoping that all the careful review is sufficient for the direct acceptance of the manuscript, thank you for your time and consideration.

Best wishes,

Mohamed. Y. M. Hanfi

on behalf of all co-authors